# Exploiting correlations in multi-coincidence Coulomb explosion patterns for differentiating molecular structures using machine learning

Anbu Selvam Venkatachalam ⓘ, Loren Greenman ⓘ, Joshua Stallbaumer, Artem Rudenko ⓘ, Daniel Rolles & Huynh Van Sa Lam ⓘ ✉

Coulomb explosion imaging (CEI) can map the real-time coordinated motion of atoms in molecules during ultrafast photochemical reactions via correlations embedded in the resulting high-dimensional data. However, this rich information remains largely underexploited due to challenges in visualizing relationships between multiple observables in multidimensional parameter space. Here, we present a new approach to CEI of polyatomic molecules, detecting up to eight ionic fragments in coincidence and leveraging machine-learning-based analysis to identify patterns and correlations. Our method yields high-dimensional, background-free momentum-space data and establishes an automated, scalable framework for extracting insightful structural information, enabling robust identification and differentiation of molecular structures. We demonstrate the method by imaging and distinguishing dichloroethylene isomers, showcasing its potential for broader applications in molecular imaging. Our results pave the way for channel-specific analysis of ultrafast structural dynamics in chemically relevant systems, particularly for disentangling mixed reaction pathways and detecting contributions from weak channels and minority species.

The ability to visualize and characterize molecular structures has long been a cornerstone of scientific discovery, enabling researchers to elucidate mechanisms of chemical reactions, design novel materials, and develop targeted therapeutics. Recent advances in ultrafast imaging techniques have revolutionized our capacity to directly observe the transformation of molecular structures during chemical reactions, shedding light on fundamental processes such as bond breaking, isomerization, and electronic excitation. These developments provide foundational knowledge spanning multiple scientific disciplines[1–8].

Coulomb explosion imaging (CEI)[9] has emerged as a powerful and promising technique for tracking time-dependent molecular motions when coupled with ultrafast light sources in a pump-probe scheme[10]. It provides excellent temporal resolution, high sensitivity to light atoms,

and direct access to three-dimensional (3D) information, even though the inversion to real-space molecular geometries is not always possible. In time-resolved CEI, the molecule of interest can be ionized using an intense laser[10–15] or X-ray[16–19] pulse, stripping away multiple electrons and leaving the molecule in highly charged states. The resulting Coulomb repulsion between positively charged fragments causes the molecule to explode, and the measured momenta of these fragments encode information about the molecular structure. CEI is particularly powerful when the probe radiation can break all the bonds and fully dissociate the molecule into atomic ions, and all resulting ions are detected in coincidence. In such cases, for every single shot, CEI yields information potentially sufficient to determine the absolute structures of polyatomic molecules, including enantiomers[12,20], and provides

James R. Macdonald Laboratory, Department of Physics, Kansas State University, Manhattan, KS, USA. ✉e-mail: huynhlam@ksu.edu

molecular frame information. However, detecting all ionic fragments resulting from the complete breakup of the molecule in coincidence is experimentally challenging. As a result, while CEI has been highly successful in imaging small molecules[10,21–31], its application to larger molecules has yielded only limited/partial structural information. Recent advances have addressed this limitation, demonstrating that CEI can image detailed 3D structures of gas-phase molecules with approximately ten atoms by leveraging coincidences from a subset of ions[14,17–19,32]. However, this method faces challenges when signals of interest are weak or contaminated by background noise.

Another current limitation of CEI applications for polyatomic molecules stems from the inherent complexity and multi-dimensionality of the data. Coincident CEI−whether performed in complete or incomplete mode−relies on analyzing the 3D momentum vectors of all detected ions. Whenever three or more ions are detected in coincidence, the number of correlated observables that could be important for characterizing structures or dynamics of interest becomes very large, particularly when both laboratory-frame and molecular-frame quantities are considered[15]. Although several efficient and standardized data representation schemes have been developed for three-particle analysis−such as Newton diagrams for visualizing momentum correlations, Dalitz plots for energy partitioning among fragments[33], and more recently, the "native-frame" approach using conjugate momenta in Jacobi coordinates[34,35]−these methods only partially capture the rich information embedded in CEI datasets even for the three-body breakups. As the number of detected fragments increases, especially when pump-probe time delays are included[18,36], the parameter space grows rapidly. Observables defined and visualized by conventional human-driven analysis typically sample only a narrow portion of this space, leaving much of the correlated structural and dynamical information unexplored.

In this work, we address these limitations by pushing CEI with kinematically complete coincidence imaging into the regime of intermediate-sized molecules. Specifically, we demonstrate that the detection of up to eight-ion coincidences is feasible with currently available tabletop laser and detector technology, extending previous reports on five-atom molecules[9,12,37–40]. In such "complete" CEI measurements, where all created ionic fragments are detected in coincidence, strict momentum conservation ensures background-free data, allowing for the unambiguous identification of weak reaction pathways[41] and contributions from minority species such as dimers in dilute samples. Imaging all atoms in a polyatomic molecule in a single shot provides extraordinarily rich structural information, opening new opportunities for investigating time-resolved structural dynamics of photoinduced reactions of chemically relevant organic molecules with unprecedented details.

As a second major step forward, we present a new analysis framework based on machine learning (ML) to help interpret the high-dimensional data from multi-coincidence CEI. Each multi-coincidence event generates a multi-vector data point (three momentum components for each fragment ion), and the distribution of such events forms a complex dataset that encodes detailed information about the structure and subtle correlations between atomic ions. As mentioned earlier, extracting meaningful insights from such data typically requires laborious analysis of the momentum distributions and manual gating on specific projections guided by human intuition, which can be challenging and prone to bias. We demonstrate that ML algorithms can efficiently recognize and exploit momentum-space patterns and correlations corresponding to distinct molecular geometries. Furthermore, we introduce a quantitative approach to determine which features in the high-dimensional CEI data are the most critical for differentiating similar structures. As molecular size increases, momentum correlations grow combinatorially more complex, making ML particularly well-suited for handling such datasets. By leveraging the readily available

ML toolbox, we establish an automated and scalable analysis framework for structural imaging using multi-coincidence CEI.

As a demonstration, we apply these advancements to imaging and differentiating isomer structures, which is a critical subject of investigation across multiple fields, including chemistry, pharmacology, biochemistry, and material science[42–45]. Although isomers share the same molecular formula, their structural differences lead to distinct physical and chemical properties that affect their behavior. For example, in pharmacology, small structural changes can result in dramatically different biological effects, as exemplified by the enantiomers of thalidomide, where one form is therapeutic and the other harmful[42]. CEI has previously been used to image chiral[12,20,46–49], geometric isomer[32,50–53], and conformer[54] configurations of molecules. In this work, we investigate the isomers of dichloroethylene (DCE). First, we present the CEI of 1,2-DCE, where the molecule fully dissociates into six atomic ions, and the full 3D momenta of all fragments are detected in coincidence. Second, using unsupervised learning, we demonstrate that coincident CEI data can be automatically separated into distinct clusters corresponding to different isomers on an event-by-event basis. Third, we employ supervised learning to determine which projections in high-dimensional CEI data are most important for distinguishing isomeric structures and similar configurations that can arise during photochemical reactions. Finally, we extend CEI to achieve up to eight-ion coincidences in isoxazole. Looking forward, the methods developed here pave the way for time-resolved investigations of larger molecular systems with all-atoms imaging and automated data interpretation.

## Results and discussion

Fig. 1a presents the ion momentum image (Newton plot) of the *cis*-DCE molecule, constructed from six-fold coincidence events in which all singly charged fragment ions (two $H^+$, two $C^+$, and two $^{35}Cl^+$) are detected. The reference frame is defined by the momenta of the two $Cl^+$ ions (see the caption for details), with the $C^+$ and $H^+$ ions plotted in this frame. A similar Newton plot for the *trans*-DCE isomer is shown in Fig. 2a. In this configuration, the momentum vectors of the two $Cl^+$ ions are nearly back-to-back, making their vector sum and consequently the $p_x p_y$ plane less well-defined (see Supplementary Fig. 1). To mitigate this, we define the $p_x p_y$ plane for *trans*-DCE using the vector difference between the momenta of the two $C^+$ ions. The resulting momentum image reveals distinct, well-separated features corresponding to each atomic fragment. This is a clear example demonstrating that one frame of reference is not necessarily suitable for all different molecular structures. One needs to combine different representations to elucidate different reaction dynamics.

The maxima in the momentum distributions of the chlorine, carbon, and hydrogen ions are well-localized, directly encoding information about the molecular geometry. Figures 1b and 2b present the results of classical Coulomb explosion simulations, assuming point charges, purely Coulombic potential, and instantaneous ionization[14,55] (see Methods for details). The simulations begin with the neutral molecule in its equilibrium geometry, with Gaussian-distributed spatial displacements and total kinetic energy (randomly partitioned among the atoms) introduced to account for the initial distribution and broadening effects due to atomic motion during the ionization process. Here, the spatial deviation of 0.25 Å and a total kinetic energy of 500 meV are used to match the width of the experimental distributions. This model successfully reproduces key features of the experimental momentum distributions, capturing the separation and localization of the fragment ions with good accuracy. This agreement suggests that the measured momentum distributions faithfully reflect the molecular structure near the equilibrium of the neutral molecule.

To provide a more quantitative comparison between experiment and simulation, Figures 1c and 2c show the azimuthal angle distributions for each ion, obtained by integrating over the radial momentum

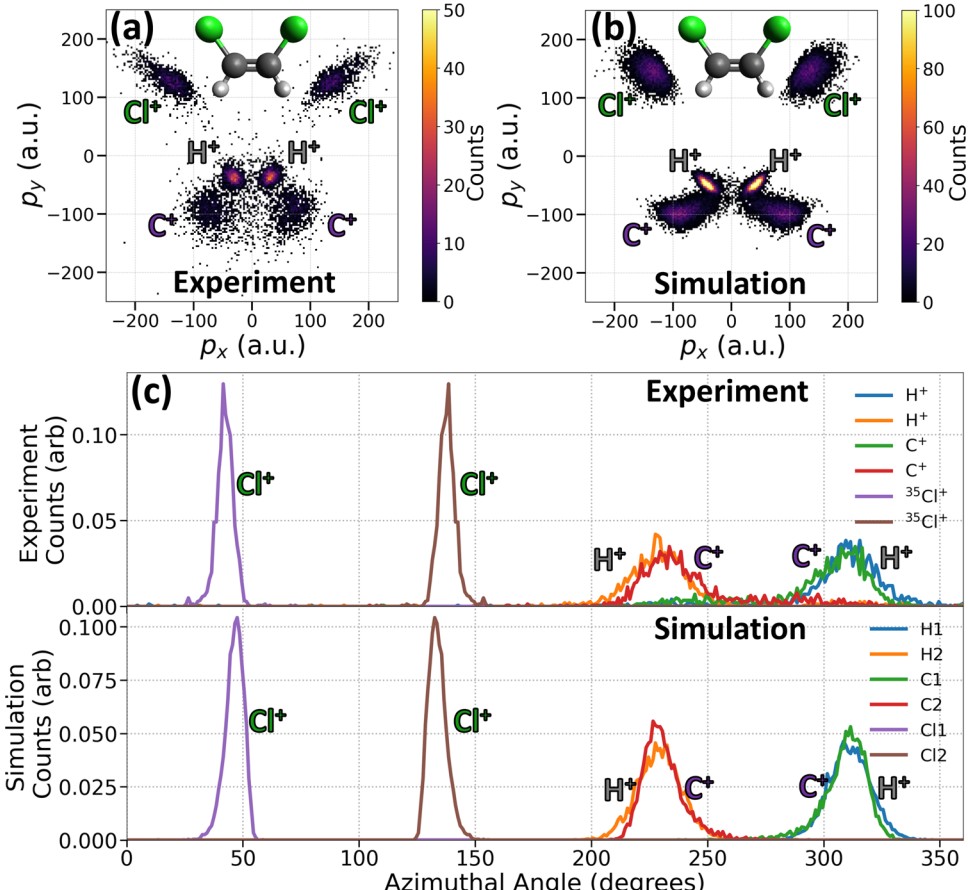

**Fig. 1 | Coulomb explosion imaging of *cis*-DCE: experiment versus simulation.**
**a** Measured and **b** simulated CEI patterns (Newton plots) of *cis*-DCE from the
(H⁺, H⁺, C⁺, C⁺, ³⁵Cl⁺, ³⁵Cl⁺) 6-fold coincidence channel. The insets show ball-and-stick
model views in the molecular plane. For each event plotted here, the coordinate
frame is rotated such that the vector difference between the two Cl⁺ momenta unit

vectors points along the $p_x$ axis and the bisector between them lies in the upper
$p_x p_y$ plane. The momenta of C⁺ and H⁺ are plotted in this coordinate frame. Panel
**c** shows the experimental (top) and simulated (bottom) distributions of azimuthal
angle for each ion (integrated over momentum magnitude). The azimuthal angle is
measured counter-clockwise from the $p_x$ axis.

coordinate. The experimental (top) and simulated (bottom) distribu-
tions exhibit excellent overall agreement, indicating that the Coulomb
explosion model effectively captures the correlated angular relation-
ships between fragment ions.

These results validate the ability of the simulation to model the
Coulomb explosion dynamics of the DCE molecules with high fidelity.
The complete coincidence detection of the full 3D momenta of all
atomic ions provides nearly background-free data, where the observed
experimental features for the 6-body coincidences are as well-defined
as those in the simulation. This level of agreement is not always
achieved in cases of incomplete coincidence detection, where
experimental distributions are broadened by contamination from false
coincidence or different final charge states (see Supplementary Fig. 2).

We also perform CEI on a sample containing a mixture of *cis* and
*trans* isomers. Fig. 3a shows the momentum pattern of this data after
rotating each event to a common frame of reference defined by the
two Cl⁺ momenta as in Fig. 1a, b. Compared to the data of only the *cis*
isomer in Fig. 1a, new features belonging to the *trans* isomer appear.
Some features are well separated from the *cis*-DCE pattern, while some
overlap. In order to automatically separate events corresponding to *cis*
and *trans* isomers from the mixture, we first perform data reduction to
reduce this data of eighteen dimensions into two dimensions, as
shown in Fig. 3b. Here, we choose to use UMAP (Uniform Manifold
Approximation and Projection)[56] —which constructs a high-
dimensional graph representation of the data based on topology and
then optimizes a low-dimensional graph to be as structurally similar as
possible— for data reduction due to its ability to handle nonlinear

patterns and its computational efficiency. A comparison between
UMAP and other popular data reduction techniques is provided in
Methods and the SM. After the dimensionality reduction, the data is
clearly separated into two groups. Events from these two groups are
correctly clustered using HDBSCAN (Hierarchical Density-Based Spa-
tial Clustering of Applications with Noise)[57] —an algorithm that builds a
hierarchy of clusters by varying the density threshold and extracts the
most stable clusters while automatically labeling low-density regions
as noise— and then colored according to their cluster labels.

We then plot the momentum images of these events separately in
Fig. 3c, d for events from clusters labeled red and blue, respectively.
The momentum image in Fig. 3c closely resembles that in Fig. 1a,
indicating that these events correspond to the *cis* isomer. Meanwhile,
the momentum image in Fig. 3d exhibits a distinct pattern that aligns
well with events from the *trans* molecules, as seen in Fig. 2a. The
excellent agreement between momentum images of events from these
two clusters and data collected with individual isomers confirms that
the data reduction and clustering algorithms above have been able to
accurately separate *cis* and *trans* isomers on an event-by-event basis,
automatically.

Now that the two isomers have been accurately clustered and
labeled, we turn to a supervised ML approach to quantitatively assess
which features contribute most to the differences between *cis* and
*trans*. The key motivation for this analysis arises from the limitations of
experimental observables, especially when used individually, in cap-
turing the structural differences between isomers. While our current
data shows a clean separation between the two isomers, it is not always

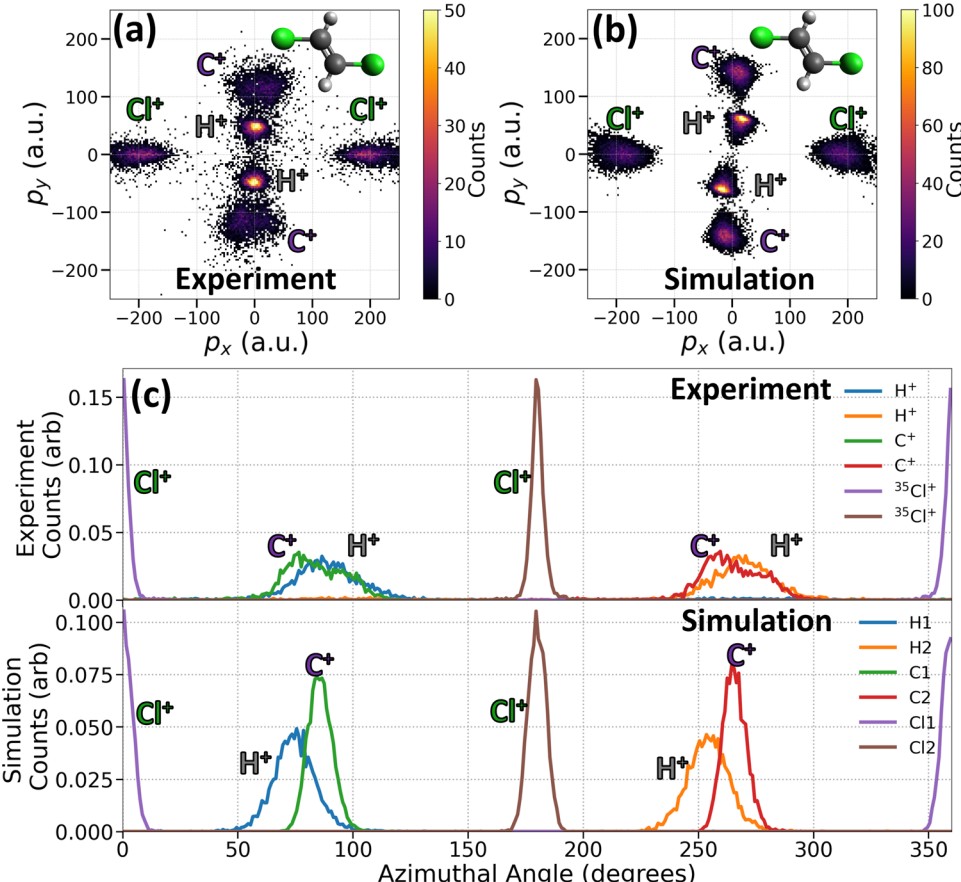

**Fig. 2 | Coulomb explosion imaging of *trans*-DCE: experiment versus simulation. a** Measured and **b** simulated Newton plots of *trans*-DCE from the ($H^+$, $H^+$, $C^+$, $C^+$, $^{35}Cl^+$, $^{35}Cl^+$) 6-fold coincidence channel. In this case, the coordinate frame for each event is aligned such that the difference vector between the two $Cl^+$ momenta is parallel to the $p_x$ axis. The $p_xp_y$ plane is established using the vector difference of the two $C^+$ momenta. The experimental (top) and simulated (bottom) azimuthal angular distributions for each ion (integrated over momentum magnitude) are given in panel **c**.

guaranteed. If many closely similar configurations coexist, isomers might appear as different parts of one big cluster, making differentiating them difficult, especially when the reduced dimension is not always readily interpretable. Our following analysis provides insights into how to construct meaningful observables to effectively differentiate similar structures. In particular, we employ the Random Forest Classifier[58]—an ensemble learning method that builds multiple decision trees using bootstrap samples and random feature subsets, then aggregates their votes to produce a more accurate and robust classification—to evaluate the discriminative power of different features. Features with high discriminative power can easily tell the two isomers apart, while ones with low discriminative power cannot cleanly separate the two. We perform this analysis for components of the momentum vectors in Cartesian coordinates and also internal momentum coordinates, such as the angle between two momentum vectors and the magnitude of the vector differences.

Fig. 4a presents the discriminative power analysis obtained from a Random Forest classifier trained to distinguish between the *cis* and *trans* isomers based on their measured Coulomb explosion momenta in the Cartesian representation [shown in Fig. 3a]. This result shows that the X and Y components of the fragment momenta are more informative than the Z component, which is expected from the planar symmetry of 1,2-DCE isomers. The effectiveness of $p_{5y}$ and $p_{6y}$ (vertical momenta of the chlorines) and $p_{1x}$ and $p_{2x}$ (horizontal momenta of the protons) in separating the isomers can be seen in Fig. 3a (and also Supplementary Fig. 16).

While the analysis in Cartesian coordinates provides insight into how momentum-space observables correlate with molecular structure, a more intuitive description that involves the momentum internal coordinates, such as $d_{ij}$ and $\theta_{ij}$, can be used. Here, $d_{ij} = |\vec{p}_j - \vec{p}_i|$ is the modulus of the difference between two momentum vectors, and $\theta_{ij} = \angle(\vec{p}_i, \vec{p}_j)$ denotes the angle between them. These features are invariant to translation and rotation, offering a robust description of the structural information independent of spatial orientation. These features have been successfully used to track changes in bond lengths[23,59,60] and bond angles[15,46] in the nuclear wave packet dynamics of molecules.

The result, shown in Fig. 4b, reveals that the angles ($\theta_{ij}$) exhibit much stronger discriminative power compared to the magnitudes ($d_{ij}$). This is because isomers have similar bond lengths, which are the main factor in determining the momentum magnitude (through the Coulomb interactions). $d_{ij}$ is more important when significant bond-length differences arise, such as during dissociation. Here, the angle correlations between fragment momenta – notably those involving pairs of $H^+$, $C^+$, and $Cl^+$ ions – serve as strong distinguishing factors between the isomers. While the role of $Cl^+$ and $H^+$ ions was evident in Cartesian coordinates, this representation highlights the significant contribution of the angle between $C^+$ fragments, providing additional structural cues for isomer differentiation.

Figure 4c shows the distribution of the angle between two $Cl^+$ fragments ($\theta_{56}$). This quantity was previously identified as the defining structural characteristic of *cis* and *trans* configurations in similar cases[50,52,53], which we confirm and quantify as the strongest single discriminator for the two isomers in our analysis. In our current data, this feature by itself can separate the two isomers without any overlap, unlike the partial overlap reported in three-body coincidence

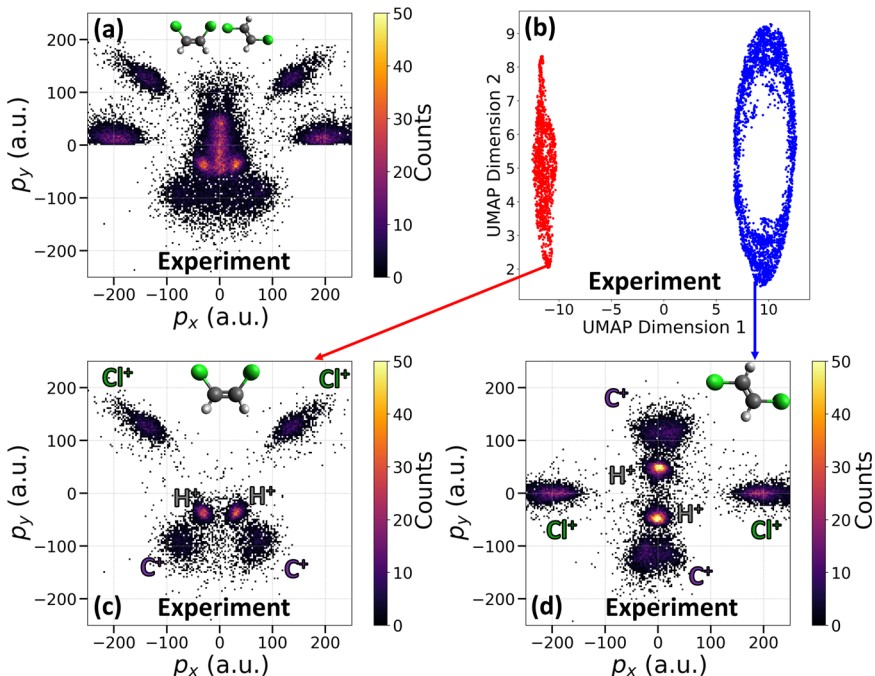

**Fig. 3 | Automatic separation of *cis* and *trans* isomers events from experimental data of a mixture. a** Newton plot of a mixture of isomers. **b** Data reduction using UMAP, where the events are colored according to their labels obtained from clustering using the HDBSCAN method. **c, d** are Newton plots of *cis* and *trans* molecules after separation, respectively. The Newton plots of the mixture (**a**) and of the *cis* isomer (**c**) use the same reference vectors as defined in Fig. 1. For the *trans* isomer in **d**, the reference frame is described in Fig. 2.

studies[50,52,53]. In Fig. 4d, we further incorporate the angle between two $C^+$ ions $\theta_{34} = \angle(C^+, C^+)$ − the second-strongest discriminator − to make a two-dimensional angle correlation plot. This plot reveals two distinct islands corresponding to the *cis* and *trans* isomers. These well separated regions demonstrate that relative fragment orientations encode key molecular characteristics and reinforce the effectiveness of these angles in differentiating structural isomers.

By leveraging ML models such as Random Forest, we can systematically identify the most informative observables for Coulomb explosion imaging studies. This approach not only enhances our ability to classify isomers but also provides a framework for feature selection in future studies of polyatomic molecular fragmentation.

We now extend our analysis to include four distinct molecular geometries: *cis*-DCE, *trans*-DCE, the *twisted* 1,2-DCE intermediate geometry, and 1,1-DCE. Their ball-and-stick models are illustrated in Fig. 5a. The twisted geometry represents a midpoint in the torsional transition between *cis* and *trans* configurations, while 1,1-DCE represents a structure where hydrogen and chlorine migrations are involved (similar to acetylene-vinylidene isomerization). Together, these geometries offer a broader perspective on conformational changes that may occur in photoinduced reaction dynamics that would be desirable to identify in a time-dependent pump-probe experiment. It is important to note that the following analysis is based on simulated data, as experimental results are not available for the transient *twisted* 1,2-DCE and 1,1-DCE. Given that our simulations closely reproduce the experimental data presented earlier, we believe that this analysis is well justified and provides meaningful insights into the structural dynamics under investigation.

We will apply both unsupervised learning (i.e., clustering) and supervised learning (i.e., classification) techniques to systematically analyze the momentum-space signatures of the isomers. Figure 5a presents the clustering results obtained through dimensionality reduction using UMAP, where all molecular configurations clearly separate into distinct clusters. These clusters can be accurately

identified by HDBSCAN. This result shows that far more detailed structural differences from CEI data can be encoded in a reduced representation.

Since the *twisted* geometry is nonplanar, the dihedral angle needs to be included to distinguish these structures in real space. We mimic the effect of this quantity in the fragment momentum space by introducing a higher-order correlation − angles between planes: $\phi_{ijkl}$ − as a structural descriptor. $\phi_{ijkl}$ is calculated from four momentum vectors where each pair − $(\vec{p}_i, \vec{p}_j)$ and $(\vec{p}_k, \vec{p}_l)$ − defines a plane. The discriminative power analysis shown in Fig. 5b shows that $\theta_{56} = \angle(Cl^+, Cl^+)$ and $\theta_{12} = \angle(H^+, H^+)$ are still among the most important discriminators.

Fig. 5c shows that the 1D distribution of $\theta_{56}$ can reveal partial separation between isomers but cannot be used as a single feature to distinguish all the isomer structures. Significant overlap persists, particularly among the *twisted* and 1,1-DCE structures, demonstrating that this observable alone does not efficiently capture the difference between *cis-trans* isomerization and other processes.

The two-dimensional correlation between $\theta_{56}$ and $\theta_{12}$, as shown in Fig. 5d, slightly enhances the separation, eliminating minor overlap and cleanly resolving *cis* and *trans* from the other two (i.e., *twisted* and 1,1-DCE). However, complete differentiation of all structures requires additional dimensions. A natural question arises: which feature is most effective for differentiating *twisted*-1,2-DCE and 1,1-DCE structures? As expected, $\phi_{1256}$ − the angle between two planes formed by protons and chlorine ions − is the most critical discriminator, which can clearly separate the two (Supplementary Fig. 17). This can be quantified by performing a similar analysis to Fig. 5b, restricted to only these two structures (Supplementary Fig. 18).

Fig. 5e shows a 3D representation incorporating $\phi_{1256}$ in addition to $\theta_{56}$ and $\theta_{12}$. This visualization reveals four distinct clusters and underscores the necessity of leveraging multiple observables to achieve a clear separation of similar molecular structures. In principle, additional dimensions can be incorporated if needed.

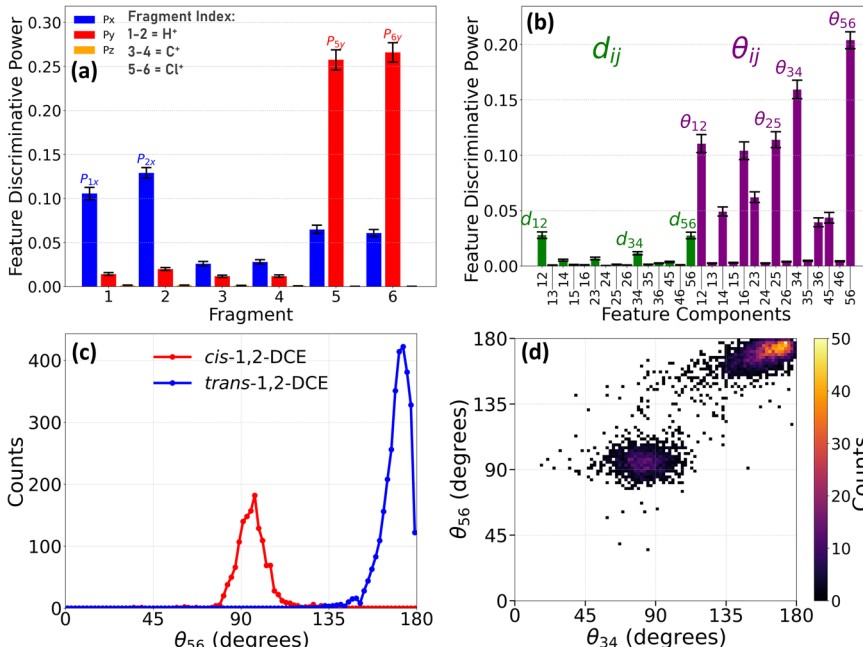

**Fig. 4 | Discriminative power analysis for distinguishing *cis*- and *trans*-DCE isomers.** Fragment indices are assigned as follows: 1–2 = H⁺, 3–4 = C⁺, and 5–6 = Cl⁺. Panel **a** highlights key features that contribute to the distinction between the two isomers for individual momentum components ($p_1$ to $p_6$ in X, Y, and Z-directions) obtained from the Random Forest classifier, where larger values indicate greater discriminative power. The error bars in panel **b** is similar to **a**, but for different features $d_{ij} = |\vec{p}_j - \vec{p}_i|$ (green) and $\theta_{ij} = \angle(\vec{p}_i, \vec{p}_j)$ (purple) (see text for more description). Error bars show the standard error of the mean of feature importance over 100 independent Random Forest fits with different random seeds. Panel **c** presents the experimental distribution of the angle between the two Cl⁺ ions for both *cis* (red) and *trans* (blue) isomers. Panel **d** is the two-dimensional angle correlation plot between $\theta_{34}$ and $\theta_{56}$.

Overall, these findings reinforce the key insight that measurement with low-dimensional data is insufficient for robust classification and highlight the advantages of the high-dimensional data provided by multi-coincident CEI. Furthermore, it is not a priori clear which observables will be most important, and the ML techniques presented here provide an automatic way of determining these quickly. Notably, this analysis does not require differentiation between ions of the same element, simplifying its practical implementation in experiments. In principle, CEI data can also be further exploited to distinguish these seemingly identical ions (of the same element), an interesting topic to be explored in a future publication.

To test the limits of the dimensionality reduction approach, we next explore how a large spread of possible product geometries affects the ability to differentiate between structures. Photoexcitation deposits substantial energy into the molecules. This additional energy can broaden their spatial and kinetic energy distributions, thereby widening the final fragment-momentum spread relative to ground-state isomers. Fig. 6a shows the simulated six-body momentum map for the (H⁺, H⁺, C⁺, C⁺, Cl⁺, Cl⁺) channel of a mixture of *cis*-, *trans*-, *twisted*-1,2-DCE, and 1,1-DCE mimicking such scenario. In this simulation, the parameters for spatial deviation and kinetic energy are 0.25 Å and 500 meV for *cis*- and *trans*-1,2-DCE (same as before), and 0.5 Å and 3 eV for *twisted*-1,2-DCE and 1,1-DCE. As expected, the momentum distribution is very broad and without any visually distinctive features that could be assigned by eye to a particular geometry. Reducing the high-dimensional data to 2D using unsupervised UMAP as before [Fig. 6b] results in partially overlapping clouds with less pronounced separation between different geometries (especially between *cis*-1,2-DCE and 1,1-DCE isomers). In this situation, one can consider looking at fragmentation channels with higher final charge states (if available), which increases the separation (Supplementary Note 4D). On the other hand, one can also use the simulated data to guide the experimental analysis. The idea is to simulate CEI of a few key geometries and perform data reduction, creating a 2D map of structures for guiding

experimental analysis. Experimental data of *cis*- and *trans*-1,2-DCE (gray) plotted on the same coordinates show very good overlap with their respective simulation clusters (colored by true labels). However, since the separation between geometries is not sufficiently distinct, automatic clustering is difficult. To overcome this, one can train a supervised UMAP embedding on the labeled simulated data to obtain better separation for automatic clustering analysis. The algorithm optimizes two nonlinear combinations of the original momentum components that maximize the separation between the four geometries, producing a 2D latent space that cleanly resolves four well-separated clusters [Fig. 6c]. Projecting the experimental data of *cis*- and *trans*-1,2-DCE into this simulation-trained latent space (gray) shows near-perfect overlap with their respective simulation clusters. This excellent agreement validates the feasibility of this approach, showing that supervised machine learning on pure simulation can serve as an appropriate guide for experimental data analysis. Finally, simple density−based clustering (HDBSCAN) of the experimental data in this supervised space recovers ≈ 99% of trans and ≈ 84% of cis events, with an overall misclassification rate of just 5.5% [Fig. 6d]. Comparable metrics are obtained when the simulation parameters are varied over physically reasonable ranges (Supplementary Note 4C), indicating that the results are robust to uncertainties in the details of the simulation. These results also show that our model can generalize well to real data. We attribute this high performance to two main factors: first is the "complete" CEI mode, which encodes very rich structural information, and second is the ability of UMAP in preserving both global (large-scale changes between different isomers) and local (small-scale variation of each isomer) structures. The unique combination of these two techniques makes the identification of molecular structures very robust. It is worth noting that, in many cases, there is no linear combination of the original momentum components that can produce a comparable separation of all four geometries, making it virtually impossible for conventional analysis to achieve the results demonstrated here (Supplementary Note 4C). In contrast, our

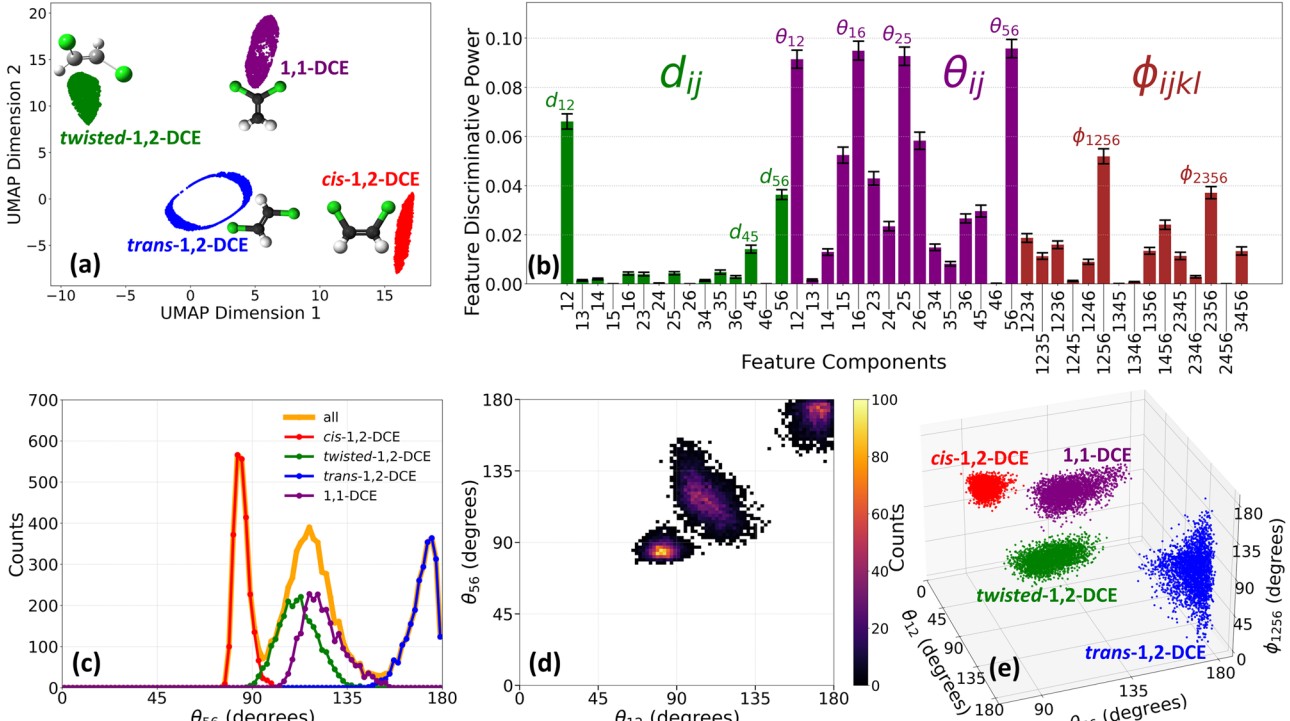

**Fig. 5 | Multidimensional analysis for structure differentiation. a** Dimensional reduction and clustering analysis of a mixture of four isomers—*cis*-, *trans*-, *twisted*-1,2-DCE, and 1,1-DCE—where ball-and-stick models of these isomers are also illustrated. **b** Discriminative power analysis of features constructed using higher-order correlations between momentum vectors, categorized into modulus difference (green) and angle (purple) between two momentum vectors, and angle between two planes (brown) formed by four momentum vectors. Error bars show the standard error of feature importance computed over 100 Random Forest fits with different random seeds (**c**–**e**) demonstrate the effectiveness of high-dimensional data in differentiating isomers, where the separation between these structures is improved sequentially from one- to two- and three-dimensions.

supervised UMAP approach cleanly resolves all geometries and transfers seamlessly to real data, opening the door to monitoring complex dynamical transformation of molecular structures in pump-probe studies of polyatomic molecules.

With currently available tabletop laser and detector technology, it is possible to achieve more than six-ion coincidence. As demonstrated in Fig. 7, we can break all the chemical bonds and completely dissociate isoxazole ($C_3H_3NO$) into atomic ions and detect all these ions in the eight-body fragmentation channel ($H^+$, $H^+$, $H^+$, $C^+$, $C^+$, $C^+$, $N^+$, $O^+$). Momentum conservation, manifesting in diagonal lines with negative slope, is indicated in the coincidence map in Fig. 7a, and the corresponding CEI pattern is shown in Fig. 7b. Previously, we have used a subset of four ions ($H^+$, $C^+$, $N^+$, $O^+$) in coincidence to create a similar image of this molecule[14]. Fig. 7c compares the distributions of azimuthal angles for the ions from the complete eight-body (solid) and the partial four-fold (dotted) coincidences. Their main features are in good agreement, similar to the results on DCE, but the complete coincidence channel shows narrower distributions, and its background-free nature can be seen, for example, in the zero baseline of the $C^+$ and $H^+$ distributions, which can be exploited to characterize contributions from weak channels and minority species. This example of eight-fold coincidence highlights the potential applications of the presented method to a broad range of molecular systems.

In conclusion, our work demonstrates the power of "complete" CEI—where all atomic ions are detected in coincidence—in providing background-free, detailed structural information of isolated, intermediate-sized polyatomic molecules on a shot-by-shot basis. We show that such complete coincident measurement of up to eight ionic fragments is feasible with regular tabletop laser sources. This capability opens the door to follow the time-dependent motion of all the atoms during molecular structural transformation in photoinduced

chemical reactions at the single-molecule level. In order to fully exploit the rich information embedded in multi-coincidence Coulomb explosion patterns, we introduce an automatic, scalable ML-based analysis framework, providing a powerful approach for identifying subtle structural variations, which was successfully demonstrated on dichloroethylene.

The method demonstrated here can facilitate the investigations of other dynamics, such as ultrafast proton transfer[61], fragmentation[62], and symmetry-breaking[63] dynamics in dimers of triatomic molecules (six atoms) or intermediate-sized molecules (up to eight atoms) with unprecedented structural insights. This framework naturally extends to larger polyatomic molecules[14,17,19,32] and can further accommodate conformers, chiral molecules, and molecular dimers, where multi-dimensional CEI combined with ML can help resolve subtle differences in fragmentation patterns between coexisting configurations. While "complete" CEI with six- and eight-ion coincidences reported in this work represents a substantial advancement compared to previous work, we anticipate a feasible extension to even higher-fold coincidence measurements for larger molecular systems in the near future by leveraging several experimental developments, including higher-repetition-rate, intense light sources (tens of kilohertz to megahertz), advanced detector technologies, and improved data analysis pipelines[19,64,65] (a comprehensive discussion is provided in Supplementary Note 1). Recent work has proposed clustering algorithms as a potential tool for distinguishing structurally similar proteins based on simulated average explosion footprints[66]. While our current work focuses on CEI, a similar ML approach can be extended to data produced by other experimental or theoretical techniques. The continued integration of advanced data science techniques into CEI and other methods will thus pave the way for more detailed and accurate imaging of molecular structures and their dynamic transformations[67–69].

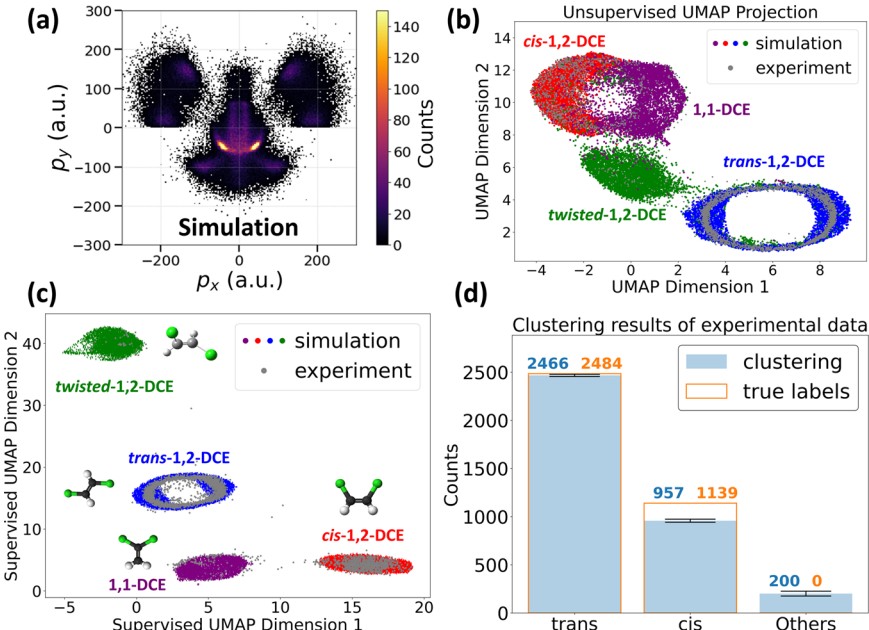

**Fig. 6 | Supervised UMAP classification of experimental CEI data. a** Simulated momentum map for the six-body fragmentation channel (H⁺, H⁺, C⁺, C⁺, Cl⁺, Cl⁺) arising from a mixture of *cis-*, *trans-*, *twisted-*1,2-DCE, and 1,1-DCE. **b** Unsupervised two-dimensional UMAP projection of the simulated events (color-coded by isomer) results in partial overlap between geometries. **c** Supervised UMAP embedding trained exclusively on the simulated labels. Experimental events (gray) are projected into the same latent space and overlap almost perfectly with their respective simulation clusters. **d** Recovery of molecular identity from the experimental data in the supervised space. Filled bars: assignment based on clustering analysis (with error bars originated from the stochastic nature of UMAP are shown in black); outlined bars: ground-truth labels derived independently. Error bars show the standard error across 100 repetitions of the supervised UMAP with different random states. Numbers above the bars denote assigned (left) and true (right) counts.

## Methods
### Experimental details
The experimental setup, shown in Fig. 8. Briefly, a Ti:sapphire laser system (Coherent Legend Elite Duo) operating at 3 kHz delivered 25-fs near-infrared pulses centered at 810 nm. The laser power was controlled using a zero-order half-wave plate and a thin-film polarizer. The pulses were focused into the interaction region of a double-sided velocity map imaging (VMI) spectrometer using a 75 mm focal-length concave mirror, reaching a peak intensity of approximately $10^{15}$ W/cm².

The molecular samples—*cis-* and *trans-*1,2-DCE (cis: ≥ 99%, Sigma-Aldrich D62209; trans: ≥ 98%, Sigma-Aldrich D62004)—were used without further purification. Due to their relatively high vapor pressures at room temperature, no heating or carrier gas was required. The sample container was connected to a stainless steel gas manifold and went through multiple freeze–pump–thaw cycles to remove air and dissolved gases. Finally, the sample vapor was expanded into a vacuum through a 30 μm nozzle into the jet chamber. A 500 μm skimmer was placed a few millimeters downstream (in the zone of silence) to select the center of the expanding molecular beam before delivering it toward the interaction region after another differential pumping stage.

Ionic fragments, produced by the interaction between the samples and the laser, are directed towards the detector using a series of electrostatic lenses, with typical voltages shown in Fig. 9.

The detector consisted of a set of 80 mm diameter microchannel plates (MCPs)—a funnel plate in front and a standard back plate—followed by a delay-line position-sensitive quad-anode (Roentdek DLD80). The funnel MCP significantly enhances the detection efficiency by widening the input area with funnel-shaped microchannels[37].

The amplified MCP and delay-line signals were processed using a constant fraction discriminator (CFD) and then recorded with a multi-hit time-to-digital converter (TDC). This setup enabled event-by-event detection of multiple coincident ions from each laser shot, similar to a COLTRIMS apparatus. For every detected ion, the time-of-flight and impact position were recorded, allowing full three-dimensional momentum reconstruction for each fragment.

In this study, we only analyzed events where all the ionic fragments were detected and discarded the rest. Specifically, we only analyzed events where we detected at least two H⁺, two C⁺ and two Cl⁺ ions for $C_2H_2Cl_2$ (6-fold coincidence) and three H⁺, three C⁺, one N⁺ and one O⁺ ions for $C_3H_3NO$ (8-fold coincidence). This was ensured by gating on the corresponding regions in the recorded ion time-of-flight mass spectrum and then applying momentum conservation constraints to reject false coincidence events, i.e., those cases where the detected ions originated from more than one molecule. After this filtering, the laboratory-frame data of the channel of interest is rotated into the recoil frame (molecular frame) as described in the main text, which allows for better data visualization and simplifies further processing since it eliminates translations and rotations from the data.

The "complete" coincidence events selected as described above constitute only a small fraction of the total measured data set. The vast majority are "incomplete" events, where one or more ions were not detected due to the finite detection efficiency, or events where the molecule did not fully atomize into singly charged atomic fragments. Our data also contains other "complete" CEI fragmentation channels (Supplementary Fig. 4), which could potentially be used to obtain a more complete picture of the molecule and its dynamics.

### Coulomb explosion simulation
Our classical Coulomb explosion simulations start with optimizing the geometry of each molecule in its neutral electronic ground state at the B3LYP/aug-cc-pVDZ level. The resulting structures are reported in Supplementary Note 3A. From this equilibrium geometry, we generated the initial condition by varying the spatial position of each atom within a Gaussian distribution of 0.25 Å standard deviation and further adding a total kinetic energy of 500 meV (randomly partitioned among the atoms), unless otherwise stated. These parameters were chosen empirically to closely reproduce the widths of the experimentally

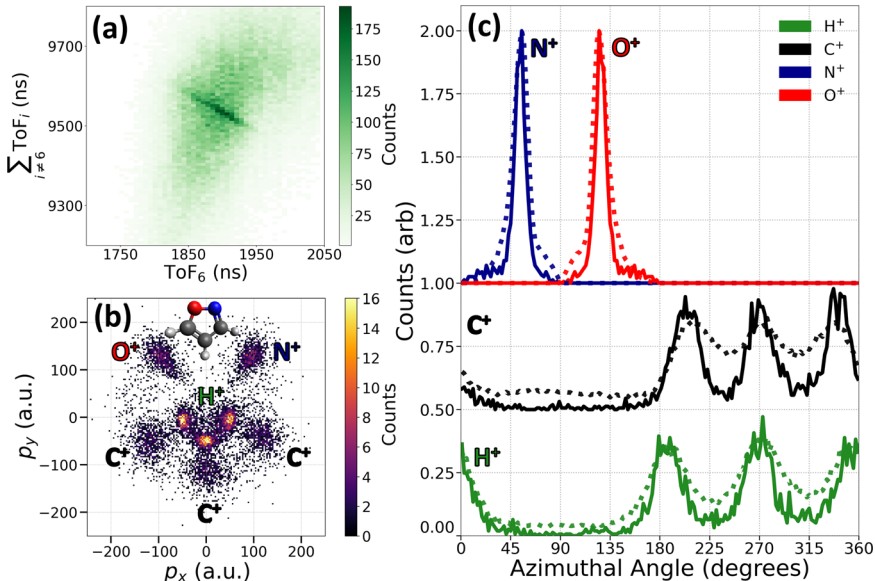

**Fig. 7 | "Complete" CEI of isoxazole with eight-ion coincidences. a** Photoion coincidence map corresponding to the eight-body fragmentation channel ($H^+$, $H^+$, $H^+$, $C^+$, $C^+$, $C^+$, $N^+$, $O^+$) of isoxazole. The coincidence ion yield is plotted as a function of the ToF of fragment 6 and the total ToF of all other ions. **b** Newton map of the eight-body complete coincidence Coulomb explosion channel. The molecular plane is defined with the vector difference between $N^+$ and $O^+$ along the $p_x$ axis

and their vector sum in the upper $p_x p_y$ plane. The inset shows a ball-and-stick model of the isoxazole molecule. **c** Distributions of the azimuthal angle for the fragment ions in the molecular plane. Solid curves represent the eight-body complete coincidence channel, while the dotted curves correspond to the four-fold ($H^+$, $C^+$, $N^+$, $O^+$) partial coincidence channel. The distribution of $C^+$ is shifted up by 0.5, $N^+$ and $O^+$ are shifted up by 1.0 for clarity.

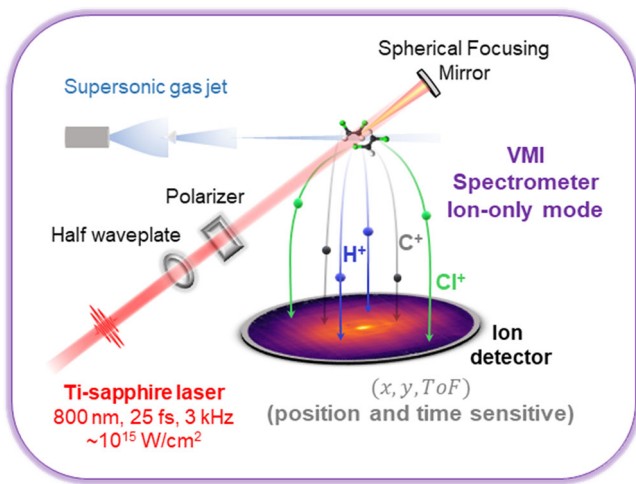

**Fig. 8 | Schematic of the experimental setup used for laser-induced Coulomb explosion imaging.** A Ti:sapphire laser (810 nm, 25 fs, 3 kHz, ~$10^{15}$ W/cm²) is focused into a cold molecular beam produced by supersonic expansion. The laser beam is directed into the interaction region using a spherical focusing mirror. Fragment ions are guided toward the detector by electrostatic fields from a double-sided velocity-map imaging (VMI) spectrometer operating in ion-only mode. The detector records the time-of-flight and impact position of all detected ions for each laser shot. This allows for coincidence detection on an event-by-event basis and enables reconstruction of the three-dimensional momenta of the fragment ions.

observed momentum distributions (as shown in Figs. 1 and 2). Although broader than the more physically meaningful Wigner distributions[14,38], this approach better captures additional broadening effects intrinsic to the Coulomb explosion process. These effects include nuclear motion during ionization, kinetic energy imparted by the laser field, and contributions from multiple ionic states, which are very challenging to compute with a fully quantum mechanical model, even for very small molecules. Our simulation agrees much better with

the experimental data compared to starting with a Wigner distribution, as shown in Supplementary Fig. 5. For statistical significance, we sample 20, 000 initial geometries per molecule. We then perform classical Coulomb explosion simulations on the sampled geometries by numerically solving coupled Newton's equations of motion, where each atom is modeled as a point charge. Our simulations assume instantaneous vertical ionization leading directly to point charges, where each atom obtains its final charge upon ionization. It also assumes that the repulsive potential of the highly charged cations leading to multibody fragmentations is purely Coulombic and that the molecule fragments completely into charged atomic ions without any internal energy.

Our simulation is equivalent to a classical molecular dynamics simulation where the force field is set as purely Coulombic interaction between point charges. The simulation is tailored specifically to the fragmentation channel of interest that we chose to investigate by filtering our coincidence data. This approach differs from more generic simulations, which aim to statistically model the distribution of multiple charge states and fragmentation pathways. Our method is thus computationally lighter and more focused, made possible by the ability to select and analyze specific channels through the "complete" coincidence detection technique.

Despite its simplicity, our Coulombic model effectively reproduces key experimental features because Coulomb repulsion significantly dominates chemical bond interactions at high-charge states in determining the fragmentation dynamics. Comparisons with a more sophisticated model using XMDYN (as demonstrated by Boll et al.[17]) show that although both models overestimate the magnitudes of fragment momenta, they accurately reproduce angle correlations between fragment momenta. Similar trends are consistently observed across various molecular systems in both laser-based and XFEL-based experiments[14,17,38], suggesting that for high charge states, the Coulomb force indeed dominates over other interactions. Thus, the simplicity of our model does not compromise the accuracy needed for clustering analyses, particularly when comparing between molecules where the angle correlation between momentum vectors is important rather

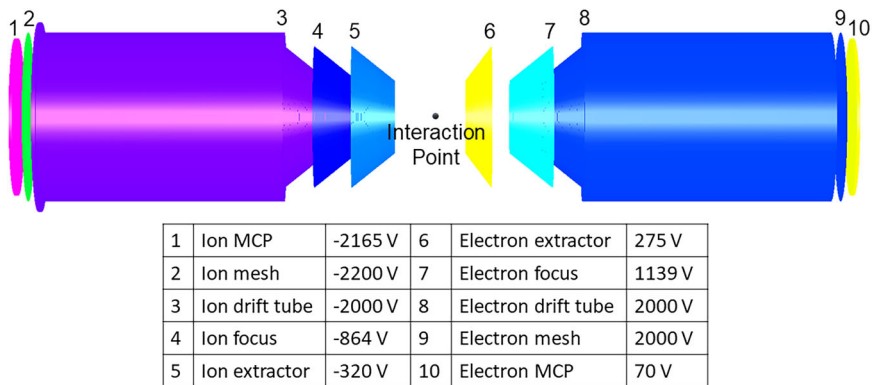

| 1 | Ion MCP | -2165 V | 6 | Electron extractor | 275 V |
|---|---------|---------|---|--------------------|-------|
| 2 | Ion mesh | -2200 V | 7 | Electron focus | 1139 V |
| 3 | Ion drift tube | -2000 V | 8 | Electron drift tube | 2000 V |
| 4 | Ion focus | -864 V | 9 | Electron mesh | 2000 V |
| 5 | Ion extractor | -320 V | 10 | Electron MCP | 70 V |

**Fig. 9 | Electrostatic layout of the spectrometer.** Schematic of the double-sided velocity-map imaging (VMI) spectrometer and the electrostatic voltages applied to each element during the experiment. The interaction region is centered between the ion-side (left) and electron-side (right) electrodes. Only the ion signals were recorded in this experiment. The voltages shown represent typical operating conditions and are listed next to their corresponding labeled elements.

than the overall absolute magnitudes. Furthermore, unlike previously demonstrated models[14,17,38] that yield overly narrow momentum distributions compared to experimental results—limiting their effectiveness in realistic clustering demonstrations—our modified model produces broader, experimentally realistic distributions (Supplementary Fig. 5). This broadening significantly enhances the practical relevance and applicability of our clustering analysis.

### Machine-learning-based analysis in Python

To analyze high-dimensional momentum-space data from multi-coincidence CEI, we employed a combination of unsupervised and supervised machine learning methods for different purposes as listed below.

- Dimensionality reduction (unsupervised): UMAP[56], Principal Component Analysis (PCA), t-distributed Stochastic Neighbor Embedding (t-SNE)[70]
- Clustering (unsupervised): HDBSCAN[57]
- Dimensional optimization (supervised): supervised UMAP, Linear Discriminant Analysis (LDA)[71]
- Feature importance ranking (supervised): Random Forest Classifier[58]

Among these techniques, UMAP (unsupervised and supervised), HDBSCAN, and Random Forest Classifier were the primary methods discussed in the main text. PCA, t-SNE, LDA are discussed in the SI for a more complete perspective, as these approaches should be used flexibly or combined as appropriate, depending on the dataset and objective. In Supplementary Note 4A, we compared the performance of different data reduction techniques on identical inputs, quantitatively quantified by computing the Silhouette Score[72] and Davies-Bouldin Index[73] for each method (more explanations in the SI). In this study, we found that UMAP consistently outperformed other methods.

Because UMAP is inherently stochastic, repeated runs on the same dataset may yield slightly different results. In Supplementary Note 4B, we evaluated the stability of our data reduction using UMAP and confirmed that the results are highly stable.

HDBSCAN was implemented using the hdbscan package[74] and used as an unsupervised clustering algorithm that identifies groups of points based on variations in local point density, without requiring the number of clusters to be specified in advance. It constructs a hierarchy of clusters using density-based connectivity, and then condenses this hierarchy to extract a flat clustering that balances stability and detail. This method was particularly effective in identifying distinct clusters in the reduced momentum-space representations of isomeric structures.

To perform supervised classification and feature importance ranking (relative discriminative power analysis), we used Random Forest Classifier from scikit-learn[75]. This ensemble method constructs a collection of decision trees using bootstrapped samples, selecting random feature subsets at each split to improve generalization. The model was trained on labeled events, and feature importance was derived from how much each feature contributed to reducing the classification uncertainty of molecular structural patterns across the ensemble of decision trees. Similar to UMAP, Random Forests are stochastic due to their initialization with pseudorandom seeds; results can vary slightly between runs. To mitigate this, we repeated the classification 100 times with different random states and reported the mean and standard deviation of the relative discriminative power.

All computations were performed on standard scientific computing hardware using free and open-source software: Python (version 3.12.3), scikit-learn (version 1.6.1), umap-learn (version 0.5.7), and hdbscan (version 0.8.39).

### Data availability

The data generated in this study are available in the Zenodo repository under https://doi.org/10.5281/zenodo.17437661.

### Code availability

Experimental data is collected using VMUSBReadout (open source, available at https://sourceforge.net/projects/nscldaq/). All machine learning analyses were performed using free and open-source software: Python (version 3.12.3; https://www.python.org/), scikit-learn (version 1.6.1; https://scikit-learn.org/), umap-learn (version 0.5.7; https://umap-learn.readthedocs.io/), and hdbscan (version 0.8.39; https://hdbscan.readthedocs.io/). The Coulomb explosion simulation code is available at https://doi.org/10.5281/zenodo.16815021.

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

## Acknowledgements

We thank Van-Hung Hoang for many useful discussions on the simulation. We are grateful to the technical staff of the J.R. Macdonald Laboratory for their support. J.S., L.G., D.R., A.R., and H.V.S.L., and the operation of the J.R. Macdonald Laboratory are supported by the Chemical Sciences, Geosciences, and Biosciences Division, Office of Basic Energy Sciences, Office of Science, U.S. Department of Energy, Grant no. DE-FG02-86ER13491. The machine learning aspect of this work was supported by a GRIPex award from Kansas State University. A.S.V. is supported by the National Science Foundation Grant No. PHYS-2409365.

## Author contributions

H.V.S.L. and D.R. conceptualized the study. A.S.V. and H.V.S.L. conducted the experiment, carried out Coulomb explosion simulations, performed machine learning analysis, and analyzed the data. H.V.S.L. and A.S.V. interpreted the results in discussion with input from J.S., L.G., D.R., and A.R. A.S.V. and H.V.S.L. produced the figures and drafted the initial manuscript. A.S.V., H.V.S.L., L.G., D.R., and A.R. contributed to iterative discussions and revisions of the final manuscript.

## Competing interests

The authors declare no competing interests.
