## [Transparent Peer Review file · Nature Communications]

Exploiting correlations in multi-coincidence Coulomb explosion patterns for differentiating molecular structures using machine learning

Corresponding Author: Professor Huynh Lam

Version 0:

Reviewer comments:

Reviewer #1

(Remarks to the Author)

This is an interesting work combining two relatively new directions in molecular spectroscopy -- the Coulomb explosion imaging (CEI) and the use of AI in linking the spectroscopic information with the molecular structure data. Taken separately, both have been explored before in a number of publications, but this particular combination appears quite novel and timely.

The underlying motivation is, that when the systems studied by CEI become more complex than the initial "showcase" systems, the amount of manual "human" analysis increases very rapidly and can become overwhelming. Thus, AI-based tools may be the answer. The authors present a well-chosen case of cis- and trans- isomers and demonstrate the application of the technique. The work is of high quality both experimentally and in regards of the machine learning (although I cannot judge the details here), and as a demonstration can be recommended for publication.

In general, it is clearly and logically written and the figures are illustrative. But some weaknesses should be reviewed by the authors:

1. The text makes it seem as if proceeding to the higher-number of coincident ions in experiment is not an obstacle with tabletop lasers etc. It does not discuss the experimental limitations in applying the "complete" CEI to intermediate-sized systems. To start with, with the detector quantum efficiency of p , the probability of detecting all ionized atoms from an N -atom system is p^N , which gets very low. So how does the AI deal with the 99.9999% of the data that is not complete? Is it discarded? Is it useful? What fraction of the total measured data was used in this study as input for AI, and what had to be discarded?
2. Some descriptions of data reduction methods for ML are too cryptic for the presumed reader of Nat Comm, e.g., in page 3. Various reduction and clustering methods are just referenced by their names and it is stated that one was found better than another. Such text is not very insightful to a reader not familiar with the methods (and perhaps not willing to read the provided references). The authors might try to add some less specific info on what methods are available, how they differ, and what guided the authors in their choices.

Reviewer #2

(Remarks to the Author)

I think this is an interesting study, and where the authors do two major things:

1. Combine VMI measurements of two 6-atom molecules with UMAP.
2. Apply ML to simulated fragmentation of several molecules of similar size.

Here are my general comments:

1 - is very nice to see applied on experimental data, but not very surprising that it works since the VMI maps from the two different molecules are rather easy to separate with the naked eye. It would be more convincing if the two molecules had

given rise to more similar VMI maps.

2 - This part of the study I find more novel, but since it is based on simulations that are done with a rather simplified physical model I have some concerns about the robustness of the results.

More specific comments:

As the reader of Nat Comm is expected to be from a wide range of research fields, I think that the manuscript would benefit from being more clear on how the experiment, the simulations, and the analysis are done. One of my main criticisms about this work is that it is poorly described what has been done. It would be impossible for even for the most knowledgeable person to reproduce the experiment, simulations and or the analysis.

Experiments: I would suggest that authors make an illustration of the setup. For somebody who is not familiar to laser induced fragmentation or VMI, it would be much easier to understand what has actually been done in the study if such illustration was accessible. How were the samples prepared, how clean were the initial solution or powder? I suggest that in addition to an illustration, that the authors also explain step by step how the experiment was done - on a level that somebody with the same setup can reproduce the experiment.

How many measurements were done?

Simulations: Even if a large part of the manuscript is based on the simulated data, the simulations are not properly explained at all. I can not say that I have any feeling for what the limitations of these simulations are, and that I think is a problem. Since the much of analysis is based on the simulations, one would expect that the simulations would be using a physical description that is as good as possible. For these type of small system, even quantum mechanical description is doable, or even classical MD using available tools Xmdyn, or MolDstruct. If the authors now decide to use a simple model with no initial charges (?), no bonds and a pure coulomb descriptions, a motivation and discussion about the limitation of the model is necessary. From the data presented it seems like the simulations done agree well with the experiments, but it is not clear how these simulations are done. Again, I think that this needs to be clear, in a way that the simulations can be reproduced. The simulations also seem to be simplified, which is not necessarily bad, but it should be discussed and reflected upon. How come a model that does not take any bonds in consideration is doing such a good job reproducing the experiments?

How are the initial atomic charges taken into consideration?

Another thing that is strange is how the starting structures are created: "a neutral molecule in its equilibrium geometry" what does that mean? At what temperature is this equilibrium structure taken? Where was it taken from or how was it created? Were the atoms neutral (like I supposed they were before the ionization in the presented simulations)? "with a Gaussian-distributed spatial displacement and total kinetic energy (...) introduced to account for the initial distribution and broadening effects..." - this sounds to me like the simulations were fitted to the experimental data, instead of using a proper physical description? What does the Gaussian-distribution correspond to in terms of temperature, and how realistic are these temperatures? My understanding is that the temperature of the molecules in the experiment is very low? Why didn't the authors try to simulate the molecules in gas phase using classical MD, which is fairly simple, to get more physical starting structures? I have to emphasize again that since the manuscript relies heavily on the simulated data, the simulations should be done properly.

What does it mean that "the spatial distribution (...) matched the width of the experimental distribution"? This is not very clear.

How many simulations were done?

It is necessary with a discussion about how the simulations affect the results in the ML and UMAP study. Which parameters are crucial for the UMAP and the ML analysis to work, and which are not. It would also be interesting to try out where the limits are, for example how much does the molecule need to be ionized for UMAP to be able to cluster properly?

Analysis: It is mentioned that PCA and t-SNE was also tested, but that UMAP was better and more computationally efficient. It would be interesting to see examples from the PCA and the t-SNE. I am also not convinced that UMAP is more efficient than PCA (as the authors state), I think that is not true. Maybe this statement could be rewritten. Also if I am not mistaken, UMAP uses a random walk in the dataset, which to my understanding means that the results can vary if UMAP is run twice on the same data set. I think the manuscript needs a discussion around this. My experience tells me that PCA would also be able to separate the data into clusters, since the VMI of the experimentally measured molecules are quite different. Using PCA would probably be more interesting since it captures the main components of the structure, and the PCA plots can be interpreted in terms of the actual structure of the molecule.

I also think that the authors need to explain what the random forest classifier does, at least in brief words. In the ML analysis the authors discuss the different X, Y and Z components. I think it would be very useful if these were illustrated in a figure somewhere. Also, in that part of the analysis, I am lacking a reasoning why the author spends so much time on it. I am a bit confused as to why it is so important.

The conclusion that a single observable is not enough for robust classification seems to disagree with recent simulation study of XFEL induced fragmentation of proteins ref 67. Could the authors comment on this?

Another main criticism that I have about the manuscript is the physical insights and description that are presented. It seems like the authors assume that the molecules are always ionized to +1 for all atoms. I am not sure about the experimental set up, but it seems unlikely that is always the case, since some molecules must be in the hit by the center of the laser beam and some on the edges, and that must affect the ionization. Maybe the experimental setup is designed in a way that this doesn't matter, but this is not clear from the text. What happens to the molecules that are ionized to +5, +4, +3, +2 or to those that are

ionized more than once per atom (+7...)? What is the probability that the molecule gets exactly +6?

Where is the limit for fragmentation?

This also relates to the choice of simulation model. I assume that if the ionization is too low, the molecule doesn't fragment, and in those cases the simulation model used probably can not reproduce the experiment?

Detailed comments:

Figures: it would be beneficial if it was more clear to the reader when the data is from simulations and from experiments. I suggest this be written out in the figures, and not only in the caption.

Fig 3. are all the dots "one" experiment/simulation, or what are the dots?

line 451: the sentence starting "Fig 6c..." is unclear. Is the discussion about one specific molecule or more than one?

line 495: mistake in the ref.

Reviewer #3

(Remarks to the Author)

The manuscript presents a new approach combining Coulomb explosion imaging (CEI), multi-coincidence measurements, and machine learning to probe molecular structure and ultrafast photochemical dynamics in strong laser fields. This work is interesting as its application of machine learning technique to analyze high-dimensional momentum-space data, enabling identification of subtle fragmentation patterns in dichloroethylene isomers, that surpass conventional manual, intuition-guided analysis. This integration of advanced data science techniques with Coulomb explosion imaging to extract more detailed and accurate information of molecular structure and their dynamic transformations is of high importance for the future development of CEI.

While the study demonstrates considerable merit, several key aspects require clarification and improvement before the manuscript can be considered for publication in NC. The following comments should be addressed.

Major comments

1. Lines 207-212

Previous studies (e.g., Refs. 14, 41) have shown that Coulomb explosion induced by strong laser fields with ~10 fs duration typically produces fragment momenta substantially lower than those calculated from neutral molecule equilibrium geometries. The authors should provide a clear explanation for the quantitative agreement in momentum magnitude observed in their work, as this apparent deviation from established work requires justification.

2. It is not easy to understand the information embedded in figure 4. It would be helpful to clarify that 1 and 2 correspond to H atoms, 3 and 4 to C atoms, and 5 and 6 to Cl atoms.

3. Regarding the discriminative power analysis of DCE isomers in Fig4a. The two C atoms are completely equivalent in the two DCE isomers, yet the discriminative power for fragments 3 and 4 in Fig 4a shows notable asymmetry. Similar discrepancies are evident in Fig4b and Fig 5b. The origin of these deviations should be explained.

4. Lines 371-374:

The analysis of twisted 1,2-DCE 374 and 1,1-DCE relies on simulations of transient states. The structures of such transient states are not well localized, leading to broader spatial and kinetic energy distributions than stable isomers. To ensure reproducibility and relevance to time-resolved experiments, the authors should specify simulation parameters, discuss how these distributions influence the results in Fig. 5, particularly for ultrafast applications.

5. The data for C₃H₃NO in Fig. 6 are weakly related to the focus of the manuscript. I suggest moving this figure to the supplementary information unless it directly supports a central claim.

6. The authors highlight the potential of their ML-based framework to extend to larger polyatomic molecules, including conformers, chiral molecules, and molecular dimers, -- an exciting prospect for advancing CEI techniques. However, several experimental challenges must be addressed to demonstrate true scalability of this approach.

(1) Total ionization efficiency: for molecules with > 10 atomic components, achieving uniform ionization of all constituent atoms becomes increasingly difficult in strong laser field. Is it possible for the proposed approach to account for the potentially biased data arising from incomplete ionization?

(2) Detection efficiency: The requirement of coincident detection of all fragment ions imposes significant practical limitations. As molecular complexity increases, the total detection efficiency may degrade rapidly. Have the authors estimated the practical limits of this approach?

(3) In large molecules, charge redistribution and its structure relaxation may occur before/during Coulomb explosion. How does this timescale affect the interpretation of results?

Addressing these points would strengthen the manuscript's applicability claims and guide future experimental designs.

Minor comments,

* Line 408, Fig 5(e) should be Fig. 5(d)

* Line 495, the number to Ref. [?] is missing.

Version 1:

Reviewer comments:

Reviewer #1

(Remarks to the Author)

The authors have addressed the criticism seriously and the manuscript is sufficiently improved to be published.

Reviewer #2

(Remarks to the Author)

The authors developed ML methods to classify Coulomb explosion ion measurements, separating different isomers from a mixture of events. They tested their method on experimental data and further benchmarked it on simulated data to understand possible future extensions of their work. The research is timely and very interesting. I recommend publication in its present form.

Reviewer #3

(Remarks to the Author)

This work presents a highly compelling integration of Coulomb explosion imaging, multi-coincidence measurements, and machine learning to probe molecular structures and ultrafast photochemical dynamics. The proposed method show potential for extension to more complex molecular systems and clusters with more than 8 constitute atoms, enabling further exploration of their structural and ultrafast dynamical properties.

In their response to the referees reports, the authors have thoroughly addressed all raised concerns and significantly enhanced the Supplementary Information. They provided sound and and convincing replies to each of my comments/questions. Moreover,beyond the complete measurements of all the final charged fragments, the authors have now included references to previous studies which successfully reconstructed molecular structure through only measuring part of the final charged fragments. These analysis methods may also be used in future machine learning models and, if implemented successfully, it could greatly reduce the challenges associated with multiple-coincidence measurements. The manuscript is now clearly written, well organized, and supported by a detailed the SI file.Given these improvements, I recommend its publication in Nature Communications.

One minor note, line 495: the reference is not converted correctly and should be revised.

Dear Editor and Referees,

We sincerely thank you for considering our manuscript and for the helpful feedback and constructive comments. In response, we have substantially revised the manuscript to provide extensive details on how the experiment, simulation, and data analysis were performed.

Notably, we have expanded the discussion of our simulation model and justified its suitability for this work. We have improved our UMAP and Random Forest Classification analysis to address the reviewers' concerns about randomness and the relation of features of symmetry-equivalent atoms. Additionally, we performed new analyses to show how the machine learning approach performs under varied simulation parameters, illustrating the robustness of the UMAP and "complete" CEI combination for molecular structure classification. We have also revised our discussion on the application of the current work to other molecular systems and provided a comprehensive discussion on how the framework could be scaled to more complex molecular systems, including our perspective on handling incomplete data.

We believe that our revisions and responses listed below have addressed all the concerns brought forward by the referees and provide sufficient justification for publication of the revised manuscript in *Nature Communications*. We reproduce the entire Referee Report comments in blue below. Our responses are given in black, and any additions/corrections to the text are denoted in red. We have also uploaded an annotated version of the revised manuscript, where modifications are indicated in blue texts. All line numbers refer to the revised manuscript.

Reviewer #1 (Remarks to the Author):

This is an interesting work combining two relatively new directions in molecular spectroscopy -- the Coulomb explosion imaging (CEI) and the use of AI in linking the spectroscopic information with the molecular structure data. Taken separately, both have been explored before in a number of publications, but this particular combination appears quite novel and timely.

The underlying motivation is, that when the systems studied by CEI become more complex than the initial "showcase" systems, the amount of manual "human" analysis increases very rapidly and can become overwhelming. Thus, AI-based tools may be the answer. The authors present a well-chosen case of cis- and trans- isomers and demonstrate the application of the technique. The work is of high quality both experimentally and in regards of the machine learning (although I cannot judge the details here), and as a demonstration can be recommended for publication.

In general, it is clearly and logically written and the figures are illustrative. But some weaknesses should be reviewed by the authors:

We thank the referee for considering our work high quality, novel, and timely, and we appreciate their recommendation for publication. This is truly encouraging for us. Please find below our point-by-point answers and revisions addressing the referee's questions and concerns.

1. The text makes it seem as if proceeding to the higher-number of coincident ions in experiment is not an obstacle with tabletop lasers etc. It does not discuss the experimental limitations in applying the "complete" CEI to intermediate-sized systems. To start with, with the detector quantum efficiency of p , the probability of detecting all ionized atoms from an N -atom system is p^N , which gets very low. So how does the AI deal with the 99.9999% of the data that is not complete? Is it discarded? Is it useful? What fraction of the total measured data was used in this study as input for AI, and what had to be discarded?

We fully agree with the referee regarding the experimental challenges associated with detecting multiple particles in coincidence. Indeed, detecting multiple coincident ions has always been a significant bottleneck in charged particle imaging, limiting CEI predominantly to two- and three-body fragmentation analysis, with only limited reports extending to four- and five-ion coincidences. We explicitly discussed these limitations in the second paragraph of our introduction and never intended to suggest that obtaining higher-number coincidence detections is straightforward with tabletop lasers. The six-fold and eight-fold coincidences demonstrated in this work therefore represent a significant advance compared to previous studies. This enables "complete" CEI, which encodes very rich structural information, of molecules containing about ~ 10 atoms or clusters of small molecules—considered "intermediate-sized" or "complex" [Boll B. *et al.*, Nat. Phys. **18**, 423 (2022), Richard B. *et al.* Science **389**, 650–654 (2025)]. We have revised the conclusion to clarify this explicitly. We also provided a comprehensive discussion on the possibility of extending the current framework to larger molecular systems in Sec. I of the updated SI.

Regarding the referee's question on the data that is not complete: Yes, in this study, we retained only events in which all the atomic fragments were detected and discard the rest. Specifically, we kept only events where we detected two H^+ , two C^+ and two Cl^+ ions for $C_2H_2Cl_2$ (six-fold coincidence) and three H^+ , three C^+ , one N^+ and one O^+ ions for C_3H_3NO (8-fold coincidence) following strict momentum conservation for kinematic completeness. These fully coincident events constitute only a very small fraction of the total measured dataset (see table below), due to the strong-field ionization mechanism favoring low charges

and challenges in detection efficiency. We note that additional “complete” CEI channels are present in the data but are not been discussed in the main text (SI, Fig. S4).

Intensity	Percentage of events in the [H ⁺ ,H ⁺ ,C ⁺ ,C ⁺ ,Cl ⁺ ,Cl ⁺] channel (momentum gated)	Percentage of events with ≥ 6 hits
1x10 ¹⁵ W/cm ²	0.0026%	0.54%
3x10 ¹⁵ W/cm ²	0.0045%	6.91%

Our work demonstrates that these high-coincidence events provide an “all-atom imaging” scheme in which each single shot carries information on all the atoms in the molecule, enabling inference of absolute molecular structure on an event-by-event basis. When coupled with supervised UMAP, these “complete” CEI channels furnish a powerful, robust framework for molecular structure classification. By contrast, other events—either incomplete breakup (e.g., two- and three-body dissociation) or incomplete detection (e.g., Fig. S2)— carry only partial structural signatures. While they are certainly useful, they require additional bespoke handling, and thus fall outside the scope of this study. In Sec. I of the updated SI, we discuss how these incomplete channels might be explored in subsequent research.

Changes to the manuscript:

- We revised the conclusion to clarify our outlook.
- We added a comprehensive discussion on the prospects of extending the current framework to larger molecular systems in Sec. I of the SI, including potential use of incomplete channels.
- We described data selection and rejection criteria in the revised Methods (Experimental details, starting from line 642).

2. Some descriptions of data reduction methods for ML are too cryptic for the presumed reader of Nat Comm, e.g., in page 3. Various reduction and clustering methods are just referenced by their names and it is stated that one was found better than another. Such text is not very insightful to a reader not familiar with the methods (and perhaps not willing to read the provided references). The authors might try to add some less specific info on what methods are available, how they differ, and what guided the authors in their choices.

2.

We appreciate this suggestion. We revised the paragraph on page 3 to provide a brief, non-technical description of UMAP for dimensionality reduction and moved the detailed comparison between UMAP with other data reduction techniques (PCA, t-SNE) to Methods (ML analysis subsection) and the SI, where we now provide results from all three methods on identical input data. We also added a concise description of the HDBSCAN clustering method in the main text (now on page 4) and additional details in Methods. In summary, the main text now explains what each method does in plain language for the readers who are less familiar with these techniques, while the SI provides side-by-side results and metrics so interested readers can inspect performance differences.

Changes to the manuscript: please see the changes on page 3 and 4. They are marked as blue text in the marked-up version of the revised manuscript. A comprehensive discussion on PCA, t-SNE and UMAP is now provided in Sec. IV A of the SI.

Reviewer #2 (Remarks to the Author):

I think this is an interesting study, and where the authors do two major things:

1. Combine VMI measurements of two 6-atom molecules with UMAP.
2. Apply ML to simulated fragmentation of several molecules of similar size.

Here are my general comments:

1 - is very nice to see applied on experimental data, but not very surprising that it works since the VMI maps from the two different molecules are rather easy to separate with the naked eye. It would be more convincing if the two molecules had given rise to more similar VMI maps.

2 - This part of the study I find more novel, but since it is based on simulations that are done with a rather simplified physical model i have some concerns about the robustness of the results.

We thank the referee for considering our work an interesting study and our experimental data analysis as a “very nice” demonstration. We addressed your specific concerns on various aspects by providing a point-by-point reply to each comment as listed below.

More specific comments:

As the reader of Nat Comm is expected to be from a wide range of research fields, i think that the manuscript would benefit from being more clear on how the experiment, the simulations, and the analysis are done. One of my main criticism about this work is that it is poorly described what has been done. It would be impossible for even for the most knowledgeable person to reproduce the experiment, simulations and or the analysis.

Experiments: I would suggest that authors makes an illustration of the setup. For somebody who is not familiar to laser induced fragmentation or VMI, it would be much easier to understand what has actually been done in the study if such illustration was accessible. How were the samples prepared, how clean were the initial solution or powder? I suggest that in addition to an illustration, that the authors also explain step by step how the experiment was done - on a level that somebody with the same setup can reproduce the experiment.

How many measurements were done?

We thank the reviewer for the suggestion. We have revised the Methods section to provide a clearer and more complete description of the experimental setup and procedures, including schematic illustrations of the experimental setup, the sample preparation process, how the data is collected and selected or discarded prior to the ML analysis. We also refer to our previous work [Lam, H.V.S. et al., Phys. Rev. Lett. **132**, 123201 (2024)] where the same experimental setup is described, which includes some additional details and specifications. This reference also includes a description of a similar simulation model, comparison with other experimental data and a discussion on its limitation. Together, we believe they provide sufficient information for reproducing the experiment by others with a comparable setup and also help readers unfamiliar with laser-induced fragmentation or VMI techniques visualize the setup. We hope that these changes substantially improve the clarity and reproducibility of the experimental methodology.

Changes to the manuscript: please see the revised Methods (Experimental details) section.

Simulations: Even if a large part of the manuscript is based on the simulated data, the simulations are not properly explained at all. I can not say that I have any feeling for what the limitations of these simulations are, and that I think is a problem. Since the much of analysis is based on the simulations, one would expect that the simulations would be using a physical description that is as good as possible. For these typ of smal system, even quantum mechanical description is doable, or even classical MD using available tools Xmdyn, or MolDstruct. If the authors now decide to use a simple model with no initial charges (?), no bonds and a pure coulomb descriptions, a motivation and discussion about the limitation of the model is necessary.

From the data presented it seems like the simulations done agree well with the experiments, but it is not clear how these simulations are done. Again, I think that this needs to be clear, in a way that the simulations can reproduced. The simulations also seems to be simplified, which is not necessarily bad, but it should be discussed and reflected upon. How come a model that does not take any bonds in consideration is doing such a good job reproducing the experiments?
How are the initial atomic charges take into consideration?

Another thing that is strange is how the starting structures are created: “a neutral molecule in its equilibirum geometry” what does that mean?

At what temperature is this equalibrium structure take? Where was it take from or how was it created? Were the atoms neutral (like I supposed they where before the ionization in the presented simulations)? “with a Gaussian-distributed spatial displacement and total kinetic energy (...) introduced to account for the initial distribution and broadening effects...” - this sounds to me like the simulations were fitted to the experimental data, instead of using a proper physical description? What does the Gaussian-distribution correspond to in therms of temperature, and how realistic are these temperatures? My understanding is that the temperature of the molecules in the experiment is very low? Why didnt the authors try to simulate the molecules in gasphase using classical MD, which is fairly simple, to get more physical starting structures? I have to emphasise again that since the manuscript relies heavily on the simulated data, the simulations should be done properly.

What does it mean that “the spatial distribution (...) matched the width of the experimental distribution”? This is not very clear.

How may simulations were done?

The referee’s questions have made us realize that our initial description lacked sufficient detail, and we believe that most of the questions from the referee originate from this lack of information. In this revision, we now provide a full Methods subsection (“Coulomb explosion simulation”) containing a comprehensive explanation and explicit justification of our simulation model, directly addressing the referee’s concerns.

Description of our simulation model: In the revised manuscript, we added a new section called “Coulomb explosion simulation” in Methods. The first two paragraphs of this section provide a description of our simulation model, which is copied below to streamline the discussion.

“Our classical Coulomb explosion simulations start with optimizing the geometry of each molecule in its neutral electronic ground state at the B3LYP/aug-cc-pVDZ level. The resulting structures are reported in Sec. III A of the SI. From this equilibrium geometry, we generated the initial condition by varying the spatial position of each atom within a Gaussian distribution of 0.25 Å standard deviation and further adding a total kinetic energy of 500 meV (randomly partitioned among the atoms), unless otherwise stated. These parameters were chosen empirically to closely reproduce the widths of the experimentally observed momentum distributions (as shown in Fig. 1 and Fig. 2). Although broader than the more physically meaningful Wigner distributions [Lam, H.V.S. *et al.* Physical Review Letters **132**, 123201 (2024)

and Bhattacharyya, S. *et al.* The Journal of Physical Chemistry Letters **13**, 5845 (2022)], this approach better captures additional broadening effects intrinsic to the Coulomb explosion process. These effects include nuclear motion during ionization, kinetic energy imparted by the laser field, and contributions from multiple ionic states, which are very challenging to compute with a fully quantum mechanical model, even for very small molecules. Our simulation agrees much better with the experimental data compared to starting with a Wigner distribution, as shown in Fig. S5. For statistical significance, we sample 20,000 initial geometries per molecule. We then perform classical Coulomb explosion simulations on the sampled geometries by numerically solving coupled Newton's equations of motion, where each atom is modeled as a point charge. Our simulations assume instantaneous vertical ionization leading directly to point charges, where each atom obtains its final charge upon ionization. It also assumes that the repulsive potential of the highly charged cations leading to multibody fragmentations is purely Coulombic and that the molecule fragments completely into charged atomic ions without any internal energy.

Our simulation is equivalent to a classical molecular dynamics simulation where the force field is set as purely Coulombic interaction between point charges. The simulation is tailored specifically to the fragmentation channel of interest that we chose to investigate. This approach differs from more generic simulations, which aim to statistically model the distribution of multiple charge states and fragmentation pathways. Our method is thus computationally lighter and more focused, made possible by the ability to select and analyze specific channels through the “complete” coincidence detection technique as described in Methods (Experimental details).”

Justification of our model

As we stated above, we purposefully employed relatively simple modeling in this work to highlight the fact that extremely computationally expensive simulations are not always necessarily needed to make sense of experimental CEI data, even if they do improve the accuracy of the modeling in cases where they are applicable (i.e. for X-ray induced Coulomb explosion imaging – we note that the XMDYN ionization model used, e.g. in Boll *et al.*, Nat. Phys. **18**, 423 (2022), is *not* applicable for strong-field induced ionization).

That said, since the referee stated that our simulations “agree well with the experiments” and questioned how such a simple model could do “such a good job reproducing the experiments”, we will try to give a few conceptual arguments to explain this “surprising” agreement.

Despite its simplicity, our Coulombic model effectively reproduces key experimental features because Coulomb repulsion significantly dominates chemical bond interactions at high-charge states in determining the fragmentation dynamics. R. Boll *et al.* [Nat. Phys. **18**, 423 (2022)] provided a comparison between a more sophisticated model using XMDYN (one of the methods the referee suggested) and the instantaneous Coulombic model (their Fig. 1, which is copied below). We can see that although the Coulombic model (diamond-shaped markers) overestimates the magnitudes of fragment momenta, it captures the angle correlations between the fragment momenta rather accurately (pretty much as good as the XMDYN model). Similar trends are consistently observed across various molecular systems in both laser-based and XFEL-based experiments [Lam, H.V.S. *et al.* Physical Review Letters **132**, 123201 (2024), Bhattacharyya, S. *et al.* The Journal of Physical Chemistry Letters **13**, 5845 (2022), Boll R. *et al.* Nat. Phys. **18**, 423 (2022)]. The fact that such a simple model achieves good agreement with both XFEL and laser Coulomb explosion data therefore suggests that for high charge states, the Coulomb force dominates over other interactions. And while the overestimation of the momentum magnitudes by the simple Coulombic model certainly needs to be addressed (e.g., if the experimental data shall be used to extract accurate bond lengths through forward modeling), it does not compromise the accuracy needed for the clustering analysis presented here since it equally affects the simulations for all isomers (please see SI, Fig. S6), particularly when comparing between molecular structures where the angular correlations are more important for distinguishing the structures than the absolute momentum magnitudes.

From: X-ray multiphoton-induced Coulomb explosion images complex single molecules

a, Newton plots of N⁺ and H⁺ (**b** and **g**) and N⁺ and C⁺ (**d** and **i**) show the measured absolute momenta following Coulomb explosion of 2-iodopyridine (C₅H₄IN) (**a**) and 2-iodopyrazine (C₄H₃IN₂) (**f**) molecules. For all the shots in which I⁺, N⁺, and at least one H⁺ and one C⁺ were detected, the coordinate frame is rotated such that the I⁺ momentum points along the z axis ($p_{I_x} = p_{I_y} = 0$) and the N⁺ momentum spans the y-z plane together with the I⁺ momentum ($p_{N_x} = 0, p_{N_y} > 0$). The momentum of the third particle is plotted in this coordinate frame. No background was subtracted. The corresponding modelling results from XMDYN are shown in **c**, **e**, **h** and **j**. The diamond-shaped markers depict the asymptotic momenta when a single positive charge is instantaneously applied to each atom.

Reproduced from *X-ray multiphoton-induced Coulomb explosion images complex single molecules* by Rebecca Boll *et al.*, *Nature Physics* **18**, 423–428 (2022), © The Author(s) 2022, licensed under CC BY 4.0 (<http://creativecommons.org/licenses/by/4.0/>).

Importantly, we would like to point out that when using a (certainly more physically meaningful) Wigner distribution to describe the width of the ground-state wave packet, the simulations (both XMDYN and the simple Coulombic model) yield a much narrower momentum distribution than what is observed experimentally (please see Fig. S5 in the SI, the figure above from [R. Boll *et al.* *Nat. Phys.* **18**, 423 (2022)] and the extensive discussion in the SM of H.V.S. Lam *et al.* *Physical Review Letters* **132**, 123201 (2024)). Since narrower distributions are easier to separate into distinct groups, we felt that applying our clustering analysis to these unrealistically narrow distributions would be less meaningful and therefore opted for the modified initial condition which yields simulated patterns that are more comparable to the actual experimental data, which, we felt, was crucial for demonstrating the practical application of the method as the clustering analysis strongly depends on distribution width and shape.

Additionally, as also mentioned below in response to other ML analysis questions, in the newly added Fig. 6 in the revised manuscript and its related discussion, we show that our experimental data overlap directly on top of the simulation in many different reduced dimension representations and under varied simulation parameters. We show that UMAP's ability in preserving both global (large-scale changes between different isomers) and local (small-scale variation of each isomer) structures in combination with "complete" CEI create a robust approach for molecular structure classification, which reduces the demand on fine-grained details in the simulation.

Changes to the manuscript:

- In the revised manuscript, we incorporated these clarifications into the newly added “Coulomb explosion imaging” section under Methods.
- Additionally, Fig. S5 in the SM provides a comparison to a simulation using a Wigner distribution as the initial condition and Fig. S6 shows that the overestimation of momentum magnitudes is systematic for all the molecules, thus, is less critical when comparing different molecules.
- Fig. 6 in the revised manuscript and its related discussion on the robustness of our analysis to varied the simulation parameters will also be mentioned in more details in other responses.

It is necessary with a discussion about how the simulations affect the results in the ML and UMAP study. Which parameters are crucial for the UMAP and the ML analyse to work, and which are not. It would also be interesting to try out where the limits are, for example how much does the molecule need to be ionized for UMAP to be able to cluster properly?

As discussed above, the width of the momentum distributions is the most critical parameter for clustering analysis. Our model with a modified initial condition produces experimentally realistic broad distributions. This significantly enhances the practical relevance of our clustering demonstration.

In the revised manuscript, we also added extensive discussions on how the UMAP and ML analyses perform under varied simulation condition, showing that the results are very robust and stable.

We also showed UMAP results for fragmentation channels with different final charge states (on both experimental and simulated data), showing that higher charge states and higher coincidence lead to better clustering. This can be explained due to (1) the ratio between the width of the momentum distribution to the momentum magnitude becomes smaller (thus producing more well-separated blobs), and (2) more particles in coincidence contain more structural information.

Changes to the manuscript:

- We added Sec. IV D in the updated SI showing UMAP results on different fragmentation channels with different final charge states.
- Please see the newly added Fig. 6 in the revised manuscript and its related discussion (blue texts in the main manuscript, Sec. IV D and Sec. IV C of the SI), where we show how the machine learning analysis perform under varied simulation parameters.

Analysis: It is mentioned that PCA and t-SNE was also tested, but that UMAP was better and more computationally efficient. It would be interesting to see examples from the PCA and the t-SNE. I am also not convinced that UMAP is more efficient than PCA (as the authors state), I think that is not true. Maybe this statement could be rewritten. Also if I am not mistaken, umap uses a random walk in the dataset, which to my understanding means that the results can vary if UMAP is ran twice on the same data set. I think the manuscript needs a discussion around this. My experience tells me that PCA would also be able to separate the data into clusters, since the VMI of the experimentally measured molecules are quite different. Using PCA would probably be more interesting since it captures the main components of the structure, and the PCA plots can be interpreted in terms of the actual structure of the molecule.

Since our previous discussion on the comparison between different data reduction techniques was, indeed, too short to provide sufficient justice, we have expanded the comparison and moved it to the SI, where more details are now given for the interested readers. Specifically, we have added the clustering

results using PCA and t-SNE on identical input data for a comparison with UMAP in Sec. IV A of the updated SI. We also used proper metrics to measure the clustering performance of each method.

In general, UMAP performs better for clustering because UMAP's nonlinear manifold approximation can uncover complex, curved structures and cluster separations (similar to t-SNE) while PCA's straight-line projections cannot capture nonlinear relationships. For large data sets with complicated structures such as proteins, the same conclusion was reached while comparing PCA and nonlinear techniques such as t-SNE or UMAP [Dorrity, M. W., *et al. Nature communications* **11**, 1537 (2020), André, T. *et al. Physical Review Letters* **134**, 128403 (2025)].

More importantly, UMAP can capture both fine-grained local connectivity and broader global trends rather than just maximizing linear variance. In the revised manuscript, the new Fig. 6 and its related discussion shows how a unique combination of this property and "complete" CEI data is particularly robust and helpful in clustering analysis for differentiating molecular structures in our case.

In practice, the actual performance depends on the specific case, of course. We agree with the referee that, in cases where the data can be separated by linear projections, PCA is preferred due to its computational efficiency, simplicity and interpretability. We have revised the text to make sure we do not inadvertently discourage the use of PCA.

The referee also raised an important question about the randomness of UMAP, i.e., the results can vary if UMAP is run twice on the same data set. This is correct. Please see Sec. IV B where we demonstrate the high stability of our UMAP results.

Changes to the manuscript:

- We added Sec. IV A of the SI where we explicitly compare different data reduction techniques with proper clustering metrics.
- We have revised the text to make sure we do not inadvertently discourage the use of PCA (sentence starting from line 756 in the main text, last sentence of Sec. IVA on page 8 of the SI).
- We added Sec. IV B of the SI where we discuss the randomness of UMAP.
- Additionally, the newly added Fig. 6 in the revised manuscript and its related discussion (blue texts in the main manuscript, Sec. IV C of the SI) how UMAP combine with "complete" CEI create a robust approach for molecular structure classification. In particular, Fig. S12 of the SI shows more comparison between UMAP and linear techniques such as PCA.

I also think that the authors need to explain what the random forest classifier does, at least in brief words. In the ML analysis the authors discuss the different X,Y and Z components. I think it would be very useful if these were illustrated in a figure somewhere. Also, in that part of the analysis, im lacking a reasoning why the author spend so much time on it. Im a bit confused as to why it is so important.

Changes to the manuscript: We added a brief explanation of Random Forest Classifier in the main text and some more details in Methods.

Regarding the ML analysis:

- The X, Y components of the fragment momenta are illustrated in many 2D plots [Fig. 1, Fig. 2, Fig. 3].
- The Z components are not important because these molecules are planar, as also demonstrated by our analysis in Fig. 4(a).
- We discuss the X, Y, Z components because Coulomb explosion data are plotted commonly in the form of momentum maps (so called Newton plot), a Cartesian coordinates representation. It is

probably the most intuitive way to visualize the data (although not necessarily the most efficient one for different purposes as also discussed in our manuscript).

- We only discuss this in Fig. 4(a) and the last paragraph of page 4, so, we hope that it does not occupy too much space of the paper.

The conclusion that a single observable is not enough for robust classification seem to disagree with recent simulation study of XFEL induced fragmentation of proteins ref 67. Could the authors comment on this?

We thank the referee for pointing out this potential confusion. Our work does not disagree with the recent simulation study by André et al. [Physical Review Letters **134**, 128403 (2025)]. Rather, our original wording might have led to ambiguity. Our intended meaning was that higher-dimensional measurements generally carry richer structural information, facilitating more effective classification compared to low-dimensional observables. For instance, a non-coincident measurement of Cl^+ ions alone from *cis*- and *trans*-1,2-DCE isomers would yield considerable overlap, thus, not efficient in differentiating different structures.

The simulation data ("explosion footprints") presented by André *et al.* are multi-dimensional, requiring at least three-dimensional reconstruction of fragment ions and a fixed molecular orientation frame, typically achieved through either three-dimensional molecular alignment or coincident detection of multiple marker ions. Even in this high-dimensional scenario, a single principal component (from PCA) or a single dimension (from t-SNE) was insufficient to separate clusters [Projection of their Fig. 3 in 1D].

Changes to the manuscript: To clarify this point and avoid misunderstanding, we have revised our manuscript by removing the phrase "single observable" and replaced it with appropriate wordings. Please see the revised sentence starting on line 435.

Another main criticism that I have about the manuscript is the physical insights and description that are presented. It seems like the authors assume that the molecules are always ionized to +1 for all atoms. I'm not sure about the experimental set up, but it seems unlikely that is always the case, since some molecules must be in the hit by the center of the laser beam and some on the edges, and that must affect the ionization. Maybe the experimental setup is design in a way that this doesn't matter, but it is not clear from the text. What happens to the molecules that are ionized to +5, +4, +3, +2 or to those that are ionized more than once per atom (+7...)? What is the probability that the molecule gets exactly +6?

Where is the limit for fragmentation?

This also relates to the choice of simulation model. I assume that if the ionization is too low, the molecule doesn't fragment, and in those cases the simulation model used probably can not reproduce the experiment?

We appreciate the referee's critique and would like to take the opportunity to clarify this important point. We do not assume that each molecule is always ionized to exactly +1 for each atom. Instead, our multi-particle coincidence measurement technique allows us to measure and select the charge of each fragment and to specifically filter out and analyze each channel of interest based on momentum conservation. In strong-field ionization, low charge states typically dominate the measured signals; however, they often contain limited or incomplete structural information. For instance, the parent ion ($\text{C}_2\text{H}_2\text{Cl}_2^+$) is identical for all DCE isomers and thus provides no distinguishing structural information. Three-body fragmentation channels such as $[\text{C}_2\text{H}_2^+, \text{Cl}^+, \text{Cl}^+]$ only carry partial structural data, omitting the arrangement of carbons and hydrogens. Our work explicitly demonstrates the feasibility of measuring six-ion coincidences for DCE molecules, providing comprehensive and rich structural insights about every atom in the molecule, which

significantly enhances the differentiation of molecular structures. We have included these clarifications in the revised Methods (Experimental Details) section.

The referee also comments that the VMI maps for *cis*- and *trans*-1,2-DCE are quite separable with the naked eye. We would like to take this opportunity to emphasize that this clean separation is a consequence of our advanced multiparticle coincidence measurements, which encode very rich information about the molecular structures. The average VMI maps without this filtering are not that “separable” (please see the revised SI, Fig. S3). Our experiments presented in this work are thus not standard VMI measurements, they are more “COLTRIMS-like” experiments [e.g., M. Pitzer, *et al.* *Science* **341**, 1096–1100 (2013), R. Boll *et al.* *Nat. Phys.* **18**, 423 (2022)] in term of coincidence detection and significantly more advanced compared to previous coincidence studies, which is one major contribution of this work.

The limit for fragmentation depends on laser intensity. As discussed in the introduction, in this study, we focus on “all-atom” imaging where the probe radiation can break all the bonds and fully dissociate the molecule into atomic ions, and all resulting ions are detected in coincidence. At lower intensities, the probability of obtaining these fully dissociated channels is negligible. Conversely, at higher intensities (around $\sim 3 \times 10^{15}$ W/cm²), we have experimentally achieved charge states as high as +14 (see SI, Fig. S4). As discussed above, higher charge states and higher coincidences lead to better clustering and differentiation between molecular structures.

Regarding the simulation, our simulation is tailored specifically to the specific fragmentation channel of interest that we choose to investigate by filtering our coincidence measurement. This approach differs from more generic simulations, which aim to statistically model distributions of multiple charge states and fragmentation pathways [as, for example, in Boll *et al.*, *Nat. Phys.* **18**, 423–428 (2022), Richard, B., *J. Phys. B: At. Mol. Opt. Phys.* **54**, 194001 (2021)]. Our method is thus computationally lighter and more focused, made possible by the ability to select and analyze specific channels through the “complete” coincidence detection technique.

Changes to the manuscript:

- We included clarifications in the revised Methods (Experimental Details) section describing on how we measured and selected the data in our analysis.
- We added Fid. S3 in the SI to show how the non-coincidence data looks like (with much less differences between the isomers), compare to how different they are after filtering the “complete” CEI channels.
- We added Fig. S4 in the SI to show exemplary “complete” CEI channels with different final charge states that we observed in our measurements.
- We added more clarification on our simulation model in Methods (Coulomb explosion simulation).
- We also added extensive discussions on the analysis when the separation in CEI patterns is not trivial in new Fig. 6 (and its related discussion in blue texts), Sec. IV C and Sec. IV D of the SI.

Detailed comments:

Figures: it would be beneficial if it was more clear to the read when the data is from simulations and from experiments. I suggest this it written out in the figures, and not only in the caption.

Changes to the manuscript: We have implemented your suggestions. Thank you!

Fig 3. are all the dots “one” experiment/simulation, or what are the dots?

Fig. 3 are all experimental data, no simulation. Each “dot” represents a six-fold coincidence event where all ions two (two H⁺, two C⁺, and two Cl⁺) were detected. All the analyses in Fig. 3 and Fig. 4 are based on experimental data.

Changes to the manuscript: We revised Fig. 3 and its caption to clarify this.

line 451: the sentence starting “Fig 6c...” is unclear. Is the discussion about one specific molecule or more than one?

Fig. 6 represents data from one specific molecule (isoxazole, C₃H₃NO). Fig. 6(c) shows a comparison between complete and incomplete coincidence detection to highlight the background-free nature of the complete coincidence measurement. Overall, Fig. 6 highlights the technical feasibility of performing high-order coincidence measurements that were not demonstrated before and thus extending the possibility of our technique to larger molecular systems.

Changes to the manuscript: Please note that Fig. 6 is now Fig. 7 in the revised manuscript.

line 495: mistake in the ref.

Changes to the manuscript: We fixed the reference. Thank you!

Reviewer #3 (Remarks to the Author):

The manuscript presents a new approach combining Coulomb explosion imaging (CEI), multi-coincidence measurements, and machine learning to probe molecular structure and ultrafast photochemical dynamics in strong laser fields. This work is interesting as its application of machine learning technique to analyze high-dimensional momentum-space data, enabling identification of subtle fragmentation patterns in dichloroethylene isomers, that surpass conventional manual, intuition guided analysis. This integration of advanced data science techniques with Coulomb explosion imaging to extract more detailed and accurate information of molecular structure and their dynamic transformations is of high importance for the future development of CEI.

While the study demonstrates considerable merit, several key aspects require clarification and improvement before the manuscript can be considered for publication in NC. The following comments should be addressed.

We thank the referee for considering our work interesting and our research topic as of “high importance for the future development of CEI”. We have addressed your specific concerns by providing a point-by-point reply to each comment as listed below.

Major comments

1. Lines 207-212

Previous studies (e.g., Refs. 14, 41) have shown that Coulomb explosion induced by strong laser fields with ~ 10 fs duration typically produces fragment momenta substantially lower than those calculated from neutral molecule equilibrium geometries. The authors should provide a clear explanation for the quantitative agreement in momentum magnitude observed in their work, as this apparent deviation from established work requires justification.

We used a similar model to the mentioned references [Lam, H.V.S. *et al.* Physical Review Letters **132**, 123201 (2024) and Bhattacharyya, S. *et al.* The Journal of Physical Chemistry Letters **13**, 5845 (2022)] with a modified initial condition. Our simulation, therefore, captures the correlations between the fragment momenta (i.e., the angles between momentum vectors) well but tends to overestimate the momentum magnitudes (or KEs) of the fragments. This overestimation of the momentum magnitudes might not be easy to see in the Newton plots (momentum image), especially when CEI data often presented as *normalized* Newton plots, where all fragment ion momenta are normalized to the momentum magnitude of the reference ion. We have used non-normalized Newton plots for a more direct comparison, but it might still not be obvious. In the revised version, we added a side-by-side comparison between the magnitude of the simulated and the experimental momenta as 1D distributions (Fig. S6) and mention this in the text (Methods) to avoid any false impressions.

While obtaining the correct magnitude of the momenta from the simulations is certainly important if one aims to determine the absolute structures of the molecules (e.g. bond lengths), our work shows that for differentiating the structures of different isomers, getting the absolute momentum magnitudes is not as crucial because the model overestimates these values for all the molecules as shown below.

Changes to the manuscript: Please see Fig. S6 in the SM and the text on line 722 (Methods, Coulomb explosion simulation), where we clarify this explicitly.

2. It is not easy to understand the information embedded in figure 4. It would be helpful to clarify that 1 and 2 correspond to H atoms, 3 and 4 to C atoms, and 5 and 6 to Cl atoms.

Changes to the manuscript: We have added labels/descriptions in Fig. 4 and its caption to clarify that 1 and 2 correspond to H atoms, 3 and 4 to C atoms, and 5 and 6 to Cl atoms according to your suggestions. Thank you!

3. Regarding the discriminative power analysis of DCE isomers in Fig4a. The two C atoms are completely equivalent in the two DCE isomers, yet the discriminative power for fragments 3 and 4 in Fig 4a shows notable asymmetry. Similar discrepancies are evident in Fig4b and Fig 5b. The origin of these deviations should be explained.

We thank the referee for this important question. We completely agree with the referee that the equivalent atoms should show similar discriminative power (within some uncertainty). After further investigation, we attribute the previous asymmetry to the stochastic nature of Random Forest Classifier (dependence on the initial random seed, called *random state*). In this revised manuscript, we run the analysis across multiple seeds and report the mean value and the standard deviation of the relative discriminative power. The updated result reflects much better the symmetry of the equivalent C atoms. Thank you!

Changes to the manuscript: Please see our revised Fig. 4(a,b) and Fig. 5(b), together with a description in Methods (Machine-learning-based analysis in Python) of how they are generated.

4. Lines 371-374:

The analysis of twisted 1,2-DCE 374 and 1,1-DCE relies on simulations of transient states. The structures of such transient states are not well localized, leading to broader spatial and kinetic energy distributions than stable isomers. To ensure reproducibility and relevance to time-resolved experiments, the authors should specify simulation parameters, discuss how these distributions influence the results in Fig. 5, particularly for ultrafast applications.

We thank the referee for this insightful question. As discussed above, the width of the final momentum distributions is a crucial variable influencing the clustering analysis. And this final width depends on the initial spatial and kinetic energy distributions of the molecules as pointed out by the referee. In this revised manuscript, we have added extensive discussions on how our analysis behaves when we sequentially broaden the initial real-space distributions and add more kinetic energy of the twisted-1,2-DCE and 1,1-DCE. As expected, the final momentum distribution of the ions become broader and the 2D reduced representation starts showing overlap between clusters. We have added two suggestions on how to deal with this situation.

- The first solution is to use the simulated data to guide the experimental analysis using supervised machine learning. The idea is to simulate CEI of a few key geometries and perform data reduction, creating a 2D map of structures. The experimental data is then plotted on this map and classified based on where the data event appears. This is demonstrated in Fig. 6 in the revised main text and Sec. IV C of the SI. We show that this method is stable and robust under varied simulation conditions.
- The second solution is to investigate fragmentation channels with higher final charge states, which will improve the separation between clusters (SI, Sec. IV D). This is demonstrated in Fig. S13, Fig. S14 and Fig. S15 of the SI. Such high charge channels can be obtained at high laser

intensity (Fig. S4) and should be easily accomplished with XFEL [R. Boll *et al.*, Nat. Phys. **18**, 423 (2022), X. Li *et al.*, Phys. Rev. Research **4**, 013029 (2022)].

In the above simulation, we have broadened the initial distributions of the twisted-1,2-DCE and 1,1-DCE in all directions. It is worth noting that the pumped molecules are often less localized than the unpumped molecules in *certain* dimensions, but not necessarily in *all* dimensions, depending on how the energy is redistributed into different vibrational modes during the photochemical reaction [e.g., Venkatachalam *et al.* arXiv:2506.07289 (2025)]. This could make the separation slightly easier.

In practice, many elements might affect the analysis. For example, if the temporal resolution is not sufficient, the observed pattern will be blurred out because geometries at different delays are convoluted at a “nominal” pump-probe delay. Similarly, a long probe pulse can also induce charge redistribution and its structure relaxation in the ionization and fragmentation process [as also pointed out by the referee in question 6 (point 3)], which further modifies the final momentum distributions, etc. We certainly do not expect to always have clean, separate clusters all the time. Our work therefore aims to go beyond this and demonstrates that (1) multi-particle coincidence measurements (especially “all-atom imaging”) encodes very rich structural information; (2) data reduction and clustering techniques such as UMAP could provide an efficient way to analyze the multidimensional data since it is designed to preserve the similarity of structures in high dimensional data into low dimensional data, meaning in the case of not-well-separated clusters, one can still partition different regions of UMAP and investigate structural changes between these regions; and (3) one can also apply Random Forest Classification analysis on the simulated data to have some guidance on what projections can give the best separation of the interested structures; and (4) one can use supervised learning (such as supervised UMAP) to create a 2D map of structures from simulated data and use this to guide the experimental analysis. Moreover, our method provides structural information on a single shot level. Hence, each data point will be considered individually without depending on the broad average wave packet (in real space). We believe that these findings are valuable for future research and of interest to a broad audience.

Changes to the manuscript:

- Please see Fig. 6 in the revised main manuscript (and its related discussion in blue texts) and Sec. IV C of the SI for supervised machine learning analysis.
- Please Sec. IV D of the updated SI for a discussion on how Coulomb explosion channels with higher charge states can improve the “resolution”.

5. The data for C₃H₃NO in Fig. 6 are weakly related to the focus of the manuscript. I suggest moving this figure to the supplementary information unless it directly supports a central claim.

In terms of machine learning analysis, we agree with the referee that the data for C₃H₃NO in Fig. 6 are weakly related to the rest of the manuscript. However, since the referee has concerns about the potential of extending this work to larger polyatomic molecules in the next question (comment 6) and Reviewer 1 also raises similar question (comment 1), we think this data (with 8-fold coincidence) is certainly important to demonstrate the technical feasibility of complete coincidence experiment on molecules with ~10 atoms. We thus prefer to keep it in the main manuscript to explicitly convey this message to the audience.

6. The authors highlight the potential of their ML-based framework to extend to larger polyatomic molecules, including conformers, chiral molecules, and molecular dimers, -- an exciting prospect for advancing CEI techniques. However, several experimental challenges must be addressed to demonstrate true scalability of this approach.

- (1) Total ionization efficiency: for molecules with > 10 atomic components, achieving uniform ionization of all constituent atoms becomes increasingly difficult in a strong laser field. Is it possible for the proposed approach to account for the potentially biased data arising from incomplete ionization?
- (2) Detection efficiency: The requirement of coincident detection of all fragment ions imposes significant practical limitations. As molecular complexity increases, the total detection efficiency may degrade rapidly. Have the authors estimated the practical limits of this approach?
- (3) In large molecules, charge redistribution and its structure relaxation may occur before/during Coulomb explosion. How does this timescale affect the interpretation of results?

Addressing these points would strengthen the manuscript's applicability claims and guide future experimental designs.

We thank the referee for asking these critical questions. In this revised manuscript, we provided a comprehensive discussion on how the current framework could be scaled to bigger molecular systems by leveraging emerging technical developments and promising data analysis approaches that can be investigated (please see Sec. I of the SI).

In practice, analyzing molecular systems containing six or eight atoms already presents significant complexity. Such analyses may involve up to 18 dimensions (even more if rotational dynamics are considered [e.g., HVS Lam *et al.*, *Phys. Rev. A* **111**, L061101 (2025)]), and commonly include secondary observables such as $\sum KE_{ij}$, ΔKE_{ij} , etc., as illustrated in many representations of three-body analyses. Thus, our new framework already offers immediate practical benefits within the demonstrated feasibility, given these challenges. We have revised the conclusion to better reflect applications of the current work (up to 8-ion coincidence) to other molecular systems and our outlook.

On Charge Redistribution and Structural Relaxation: We agree that charge redistribution and structural relaxation prior to and during Coulomb explosion are important to be considered. Recent AIMD simulations for CEI studies [e.g., Xu *et al.*, *J. Chem. Phys.* **162**, 114311 (2025)] have demonstrated fairly good agreement with experimental data, including the absolute values of the kinetic energies (albeit, for a small molecule and still narrower momentum distribution). Since this simulation assumed an instantaneous vertical transition to the electronic ground state of the highly charged states, the agreement suggests that propagation on the final charged-state potential energy surface could be the critical step, while other interactions might be less important. This still involves redistribution of charges from the initial geometry (where atoms share electrons) to the final charges where all the atoms are far from each other (all atoms with +1 charges in our case), which is missing in our current simulation. Since the interaction is not purely Coulombic at small distances, our model overestimates the absolute momenta of the fragments. However, for highly charged states, the atoms in the molecule quickly move away from each other, giving our simulation the ability to reproduce other key features in experimental data. Extending the time-dependent quantum mechanical coupled nuclear-electronic calculation to larger molecules (more degrees of freedom), especially if one aims to also include excited states (of highly charged surfaces), still poses significant challenges. This work is still in progress and hopefully will be reported in subsequent publications. For the current simulation, we have modified our initial condition in real space to be much wider than the Wigner distribution to empirically account for these broadening effects. And our work shows that this is a reasonably good model in our situation. Experimentally, shorter pulses and inner-shell ionization offer practical means to minimize these relaxation effects by rapidly triggering ionization and explosion processes on very high charge states.

We hope that the revised and added discussions have clarified and strengthened our manuscript's claims on its applicability and can guide future experimental designs as the referee suggested.

Changes to the manuscript:

- Please see Sec. I of the SI for an extensive discussion on the possibility of extending our current framework to larger molecular systems.
- We have revised the conclusion to better reflect applications of the current work (up to 8-ion coincidence) to other molecular systems and our outlook.

Minor comments,

* Line 408, Fig 5(e) should be Fig. 5(d)

This has been fixed. Thank you!

* Line 495, the number to Ref. [?] is missing.

We fixed the reference. Thank you!